# VCP-dependent muscle degeneration is linked to defects in a dynamic tubular lysosomal network in vivo

**Alyssa E Johnson, Huidy Shu, Anna G Hauswirth, Amy Tong, Graeme W Davis\***

Department of Biochemistry and Biophysics, University of California, San Francisco, San Francisco, United States

**Abstract** Lysosomes are classically viewed as vesicular structures to which cargos are delivered for degradation. Here, we identify a network of dynamic, tubular lysosomes that extends throughout *Drosophila* muscle, in vivo. Live imaging reveals that autophagosomes merge with tubular lysosomes and that lysosomal membranes undergo extension, retraction, fusion and fission. The dynamics and integrity of this tubular lysosomal network requires VCP, an AAA-ATPase that, when mutated, causes degenerative diseases of muscle, bone and neurons. We show that human VCP rescues the defects caused by loss of *Drosophila* VCP and overexpression of disease relevant VCP transgenes dismantles tubular lysosomes, linking tubular lysosome dysfunction to human VCP-related diseases. Finally, disruption of tubular lysosomes correlates with impaired autophagosome-lysosome fusion, increased cytoplasmic poly-ubiquitin aggregates, lipofuscin material, damaged mitochondria and impaired muscle function. We propose that VCP sustains sarcoplasmic proteostasis, in part, by controlling the integrity of a dynamic tubular lysosomal network.

**\*For correspondence:** graeme.davis@ucsf.edu

## Introduction

Valosin-containing protein (VCP), the homologue of yeast Cdc48, is the causative gene for a multisystem degenerative disease that was originally termed IBMPFD to encompass the wide range of debilitating clinical outcomes, including inclusion body myopathy (IBM), Paget's disease of the bone (PDB) and frontotemporal dementia (FD) (*Watts et al., 2004*). Recently, the list of degenerative disorders that are associated with VCP mutations has expanded to include amyotrophic lateral sclerosis (ALS) (*Abramzon et al., 2012*), spastic paraplegia (*Clemen et al., 2010*), scapuloperoneal muscular dystrophy (*Liewluck et al., 2014*) and Charcot-Marie-Tooth disease (*Gonzalez et al., 2014*). Currently, there are no viable treatments available to slow or halt progression of VCP-related diseases.

Muscle weakness is the first presenting symptom in over 50% of VCP disease patients (*Weihl et al., 2009*), yet very little is known about the muscle pathology of VCP-related diseases. Muscle biopsies from patients with VCP-related diseases display an accumulation of cytoplasmic poly-ubiquitin aggregates (*Watts et al., 2004*; *Weihl et al., 2009*; *Dolan et al., 2011*), suggesting a major defect in protein clearance. VCP is an AAA-ATPase that has essential functions in ubiquitin-dependent proteolysis. But, pathogenic mutations in VCP do not seem to impair the UPS or ERAD protein degradation pathways (*Tresse et al., 2010a*; *Chang et al., 2011*). More recently, VCP has been implicated in autophagy. Specifically, over-expression of VCP mutant transgenes with disease causing mutations leads to an accumulation of autophagosomes (*Ju et al., 2009*; *Tresse et al., 2010a*), suggesting that VCP functions in processes related to the maturation or fusion of autophagosomes with lysosomes.

Lysosomes are the major cellular degradation sites for clearing damaged proteins and organelles. Lysosomes are classically thought to be vesicular organelles, where they serve as depots for cargo

**eLife digest** Mutations in a gene that produces a protein called Valosin-containing protein (VCP for short) causes degenerative diseases that affect the brain, muscle and bone. In nearly half of the individuals with these VCP-related diseases—which can also result in dementia, Paget's disease of the bone and amyotrophic lateral sclerosis (ALS)—the first symptom is muscle weakness. Currently, very little is known about how VCP affects muscles.

Patients with VCP-related diseases often have problems clearing damaged proteins from their cells, and recent research suggests that VCP is important for forming a cellular structure known as a lysosome. Lysosomes contain powerful enzymes that destroy damaged proteins and other cellular structures that would otherwise accumulate in the cells. In most cells, lysosomes look like bubble-like compartments called vesicles. However, in some types of cells lysosomes have been observed to form a network of tubules that extend throughout the cell interior. However, it remains unclear what these tubules do, how they form in cells and whether they are altered in disease.

Johnson et al. analyzed lysosomes in the muscle of the fruit fly species *Drosophila melanogaster* and discovered that lysosomes were in the form of a network of tubules that spread throughout each muscle cell. These tubules constantly changed in living muscles; extending, retracting, breaking and merging to form a large tubular lysosome network. When Johnson et al. reduced the amount of VCP produced by the muscle cells, via a method called RNA interference, the lysosome tubules broke down into vesicles that were no longer constantly changing. Modifying these defective fly muscle cells so that they produced the human VCP protein caused the tubules to form again. These results suggest that the human and fly VCP proteins are very similar and that they play a key role in either the ability of lysosomes to form tubules or the maintenance of existing tubules.

Johnson et al. then engineered flies to produce a version of the VCP protein that had mutations commonly seen in individuals with degenerative diseases. Lysosome tubules did not form correctly in the muscle cells of these flies. These flies also had other abnormalities; for example, their cells showed a great build-up of damaged proteins, and their ability to move their muscles was weaker.

These findings suggest that a network of lysosomal tubules is necessary for healthy muscle cells, but how and why these tubular networks are formed or maintained is still mysterious. What causes lysosomal membranes to form tubules? How do they break and fuse? And why are they necessary? Genetic experiments in fruit flies will be a great place to discover these mechanisms and understand the links to degenerative diseases in humans.

delivered via endosomes or autophagosomes. However, in certain systems lysosomes have been observed to adopt non-vesicular morphologies. A particularly dramatic example has been observed in a subset of bone-derived cultured macrophages, where lysosomes form abundant, extended tubules that radiate from the cell center and, in some cases, form an interconnected web throughout the cytoplasm (*Swanson et al., 1987a*; *Knapp and Swanson, 1990*). Additionally, there are examples where cellular stress, particularly the induction of high levels of autophagy, induces lysosomal membranes to tubulate and undergo scission to produce new vesicular lysosomes, a process referred to as autophagic lysosome reformation (ALR) (*Yu et al., 2010*). Despite these observations, lysosome tubules have received little attention and it remains unclear to what extent lysosome tubulation occurs in different cell types and what purpose it serves in vivo. Moreover, the molecular repertoire of factors required for lysosome tubule formation is virtually unknown.

Here, we employ fluorescently tagged lysosomal and autophagic markers to study the autophagy-lysosome system in *Drosophila* muscle cells and investigate the muscle pathology of VCP-related diseases. Remarkably, we find that lysosomes adopt a dynamic, tubular morphology that ramifies throughout the entire sarcoplasm of *Drosophila* muscle, in vivo. We find that VCP is required for the integrity and dynamics of this tubular network. Disruption of lysosome tubules correlates with defects in autophagosome-lysosome fusion, increased poly-ubiquitin aggregates and the accumulation of lipofuscin material in the sarcoplasm. We show that the human VCP homolog can rescue lysosomal tubulation following loss of VCP in *Drosophila* muscle, indicating that the functions of VCP in lysosome tubulation are conserved. Finally, we demonstrate that homologous mutations that cause VCP diseases in human patients disrupt the lysosome tubular lattice, suggesting that disruption of

lysosome tubules contributes to VCP mutant pathogenesis. Taken together, our data establish a functional link between lysosome tubule dysfunction and the pathology of VCP-related degenerative diseases.

## Results

### *Drosophila* sarcoplasmic lysosomes form an extended dynamic tubular array in vivo

To visualize muscle lysosomes in vivo, we expressed RFP-tagged Spinster, which has previously been defined as a late endosomal/lysosomal transmembrane protein (*Sweeney and Davis, 2002*; *Dermaut et al., 2005*). Remarkably, when Spinster-RFP is expressed in *Drosophila* muscle, Spin-RFP localizes to an expansive tubular network (*Figure 1A–C* and *Video 1*). Tubules were found evenly distributed throughout the sarcoplasm and formed a web of connections with other tubules (*Figure 1B,C*). Tubules were observed in every muscle and there were no apparent differences in the tubule abundance or architecture between different muscles. We also observed enlarged vesicular compartments at tubule intersections throughout the muscle (*Figure 1C*). This network is highly sensitive to all chemical fixation conditions that we have attempted. When subjected to fixation, the tubule lattice collapses (*Figure 1D*), leaving behind distributed round, Spinster-positive compartments that resemble classically defined late-endosomal, lysosomal structures (*Sweeney and Davis, 2002*; *Dermaut et al., 2005*). Thus, live imaging is essential to study the function and relevance of this Spin-positive, tubular network.

We next explored the dynamics of this tubular network in time-lapse videos. We find evidence of tubule extension, retraction, scission and fusion (*Figure 1E–G*). The dominant activity in the network is the dynamic extension and retraction of individual tubules throughout the sarcoplasm (*Figure 1E* and *Video 2*). Often the same tubule was observed to extend and retract, while surrounding tubules remained constant. The impression is that the lattice has both stable, interconnected tubules and dynamic elements that create new connections or retract once a connection is broken. In less frequent instances, we observed scission events from the end of a tubule, resulting in a mobile vesicle (*Figure 1F*). In a corollary phenomenon, we observed tubule extensions that resulted in tubules fusing to larger nodes in the network (*Figure 1G*). Finally, we observed de novo formation of tubules extruding from the side of existing tubules (*Figure 1G*, last panel) demonstrating that tubulation can originate from existing tubules, not just pre-existing nodes. Importantly, the existence of a tubular network and tubule dynamics are not an artifact of a dissected neuromuscular preparation. We observed an identical, dynamic tubular lysosomal network in intact larvae that were imaged through the cuticle while restrained in a microfluidics chamber (*Video 3*).

The observed lysosomal network structure and dynamics are consistent with the involvement of either the actin or microtubule cytoskeletons. First, we tested whether the tubules require an intact microtubule cytoskeleton. Treatment with nocodazole for 1 hr completely abolished tubules, indicating that the tubules require microtubule support (*Figure 1H,I*). In contrast, disruption of the actin cytoskeleton with latrunculin A did not have a significant effect on the tubular network, indicating a non-essential role for the actin cytoskeleton in maintaining lysosome tubules (*Figure 1J*). Finally, we tested whether Clathrin is essential for tubular network integrity. Clathrin has the capacity to shape membranes and was recently implicated in the process of ALR in cultured mammalian cells (*Rong et al., 2012*), a process that involves limited tubulation from auto-lysosomal compartments. Expression of Clathrin heavy chain RNAi (*Chc-RNAi*) completely disrupted network integrity (*Figure 1K*).

Although Spinster was characterized previously as a lysosomal marker, the tubular structures labeled by Spinster are dramatically different from the classical view of vesicular lysosomes. Therefore, we performed additional experiments to validate that the tubular network is lysosomal. First, we co-imaged Spin-GFP with the low pH fluorescent probe Lysotracker and found complete co-localization (*Figure 2A*), indicating that the Spinster network is acidic. To verify that the observed tubular network is not an artifact of Spin-RFP over-expression, we stained wild type muscles with Lysotracker to examine lysosome morphology under wild type conditions. Lysotracker staining confirmed the existence of this tubular network in wild type muscle (*Figure 2B*). We also note that the tubule intersections stained more intensely for Lysotracker than the tubules themselves. This could be due to increased tubule volume at the intersections, or these sites might actually have a lower pH than the

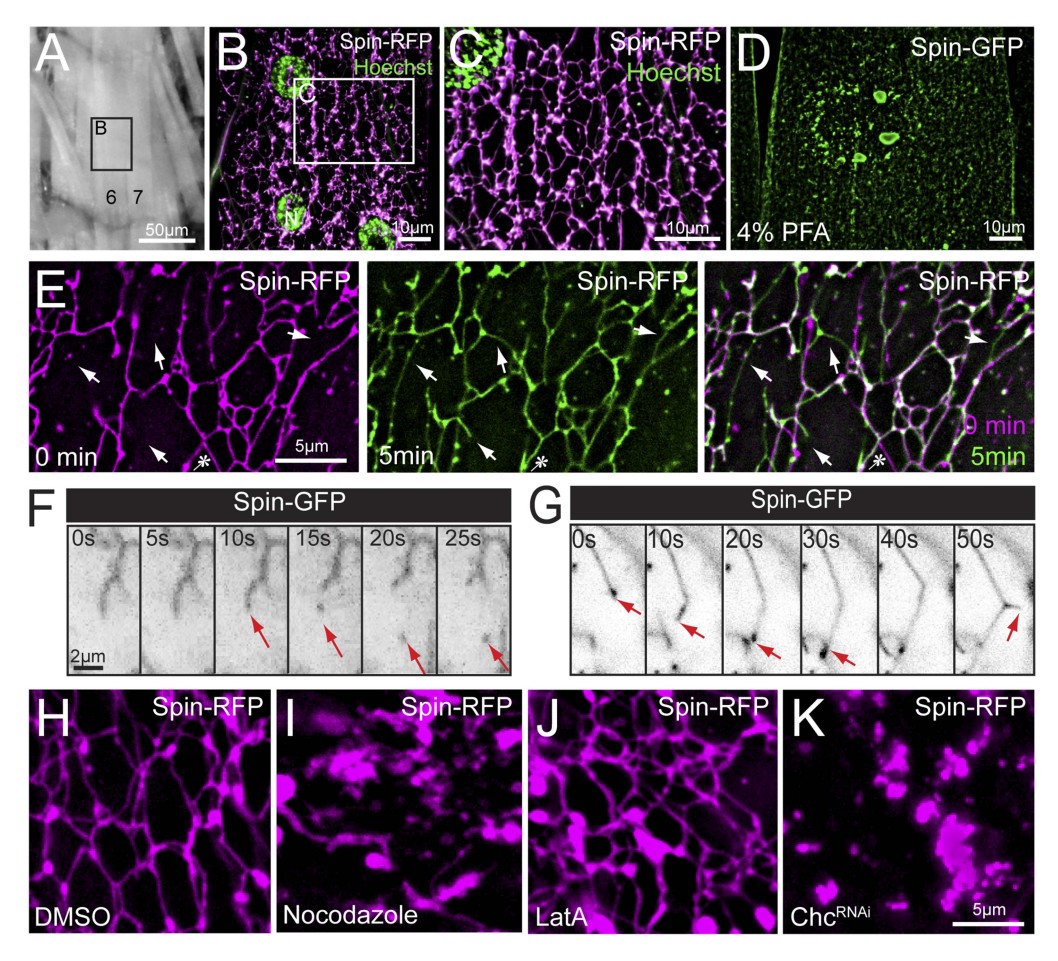

**Figure 1**. Lysosomes adopt an extended dynamic tubular array in *Drosophila* sarcoplasms. (**A**) Muscles of third instar larvae from segment A2. (**B**) Representative live image of Spin-RFP expressed in muscles at 63x magnification. Muscle 4 (**B**) is shown. DNA was stained with Hoescht. (**C**) Representative live image of *Spin-RFP* expressed in muscles using the muscle-specific *MHC-Gal4* driver. DNA was stained with Hoescht. (**D**) Representative image of Spin-GFP localization in a muscle that was fixed with 4% PFA prior to imaging. (**E**) Time-lapse images of the Spin-RFP network. Time 0 is represented in magenta, and the 5 min time-point is represented in green. The 2 time points were merged to show new and lost tubule formations over the course of 5 min. Arrows indicate examples of de novo tubule formations and the asterisk indicates a retracted tubule. (**F**) Representative time-lapse sequence of a Spin-GFP tubule fission event. (**G**) Representative time-lapse sequence of a Spin-GFP tubule fusion event. In the last frame, a de novo tubule can be seen extruding from the middle of a pre-existing tubule. (**H–J**) Spin-RFP localization in muscles treated with DMSO (**H**), Nocodazole (**I**) or LatA (**J**). (**K**) Spin-RFP localization in muscles expressing Clathrin heavy chain (Chc) RNAi.

tubules themselves. Finally, we co-imaged Spin-RFP with several other organelle markers to verify that Spinster specifically labels lysosomes and does not co-stain other organelles. ER and mitochondria also form tubule structures, but when we co-imaged Spin-RFP with ER-tracker and Mito-tracker fluorescent dyes, Spin-RFP tubules did not co-localize with either ER or mitochondria tubules (*Figure 2C,D*). Instead, Spin-RFP tubules were interwoven between mitochondria and ER tubules. Additionally, Spin-RFP did not co-localize with markers for early endosomes (YFP-Rab5), recycling endosomes (Rab11-GFP), medial Golgi (ManII-GFP) or trans Golgi (GalT-YFP) (*Figure 2E–H*). We note that Golgi organization in muscles forms vesicular structures rather than the classical Golgi stacks that are observed in most cell types and this is consistent with Golgi organization that has been observed in vertebrate skeletal muscles (*Ralston et al., 2001*). Collectively, we have identified and characterized

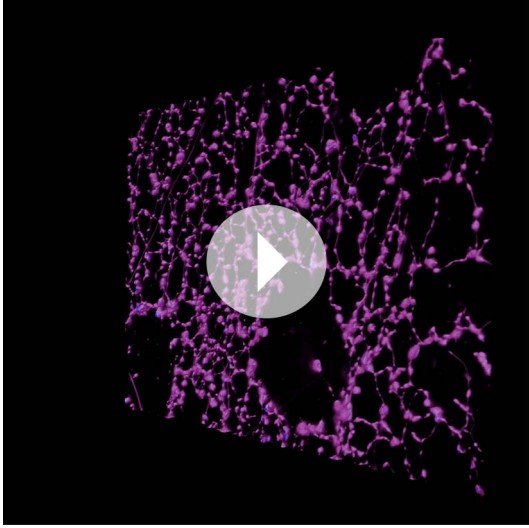

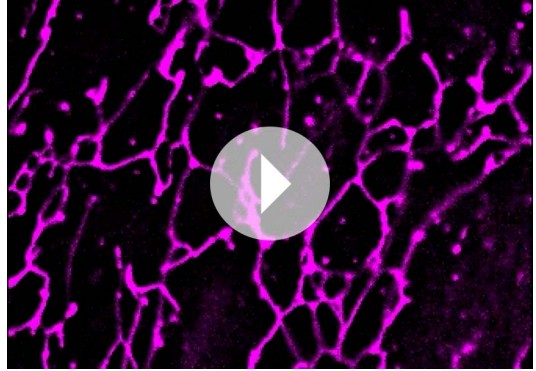

**Video 2.** Spin-RFP tubule dynamics in *Drosophila* muscle. Representative time-lapse video of Spin-RFP expressed in muscles. Frames were taken at 10 s intervals.

**Video 1.** Spin-RFP tubular network in *Drosophila* muscle. Spin-RFP was expressed in muscles and imaged live. Average Z-stacks were assembled to produce a 3D volume projection and various angles of the projections are shown.

a dynamic tubular lysosomal network that permeates the entire sarcoplasm of *Drosophila* body-wall muscle in vivo.

## VCP is required for the integrity and dynamics of the extended tubular lysosomal network

To investigate the molecular underpinnings of the observed lysosomal tubule dynamics, we pursued a candidate-based RNAi screen to identify genes required for lysosome tubulation. We focused on genes that have been implicated in the autophagy-lysosome system and identified the AAA-ATPase VCP as being required for the integrity of the entire tubular lysosome network. Specifically, inhibiting *VCP* expression by RNAi abolished lysosome tubules, leaving behind vesicles throughout the sarcoplasm (*Figure 3A,B*). The lysosome vesicles were irregular in their size and shape and appeared clustered, rather than uniformly distributed throughout the sarcoplasm. To determine whether the catalytic ATPase function of VCP is required for the tubular network integrity, we employed a VCP-selective inhibitor DBeQ (*Chou et al., 2011*). Acute inhibition of VCP with DBeQ completely disrupted the tubular network after 3 hr (*Figure 3C,D*), demonstrating the required catalytic function of VCP. Furthermore, time-lapse imaging of Spin-GFP after treatment with DBeQ for 4 hr revealed that the remaining vesicular lysosome structures are completely static (*Video 4*).

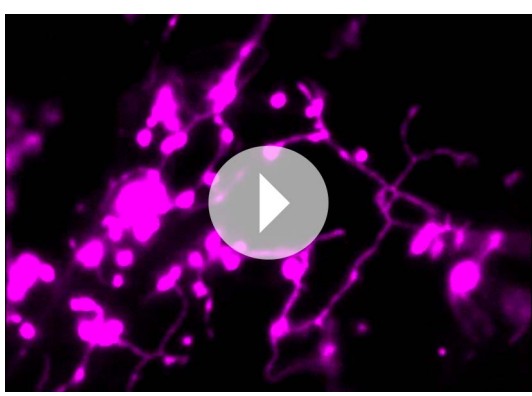

**Video 3.** Spin-RFP tubule dynamics in an intact larva. A whole un-dissected larva was immobilized in a mircro-fluidics chamber and Spin-RFP was imaged in the body-wall muscle through the transparent cuticle. Frames were taken at 10 s intervals.

To determine if the role of *VCP* in maintaining the tubular lysosomal network in muscle cells is conserved, we overexpressed human *VCP* in *dVCP^RNAi* muscles. Human *VCP* should be completely resistant to *dVCP^RNAi* due to lack of extended stretches of nucleotide identity. Overexpressing human *VCP* in *dVCP^RNAi* muscles rescued the formation of lysosome tubules in every muscle (*Figure 3E–G*). These rescue data confirm that *VCP* knockdown is the cause of disrupted lysosomal network integrity and demonstrate that VCP-dependent activity is conserved in the human VCP protein.

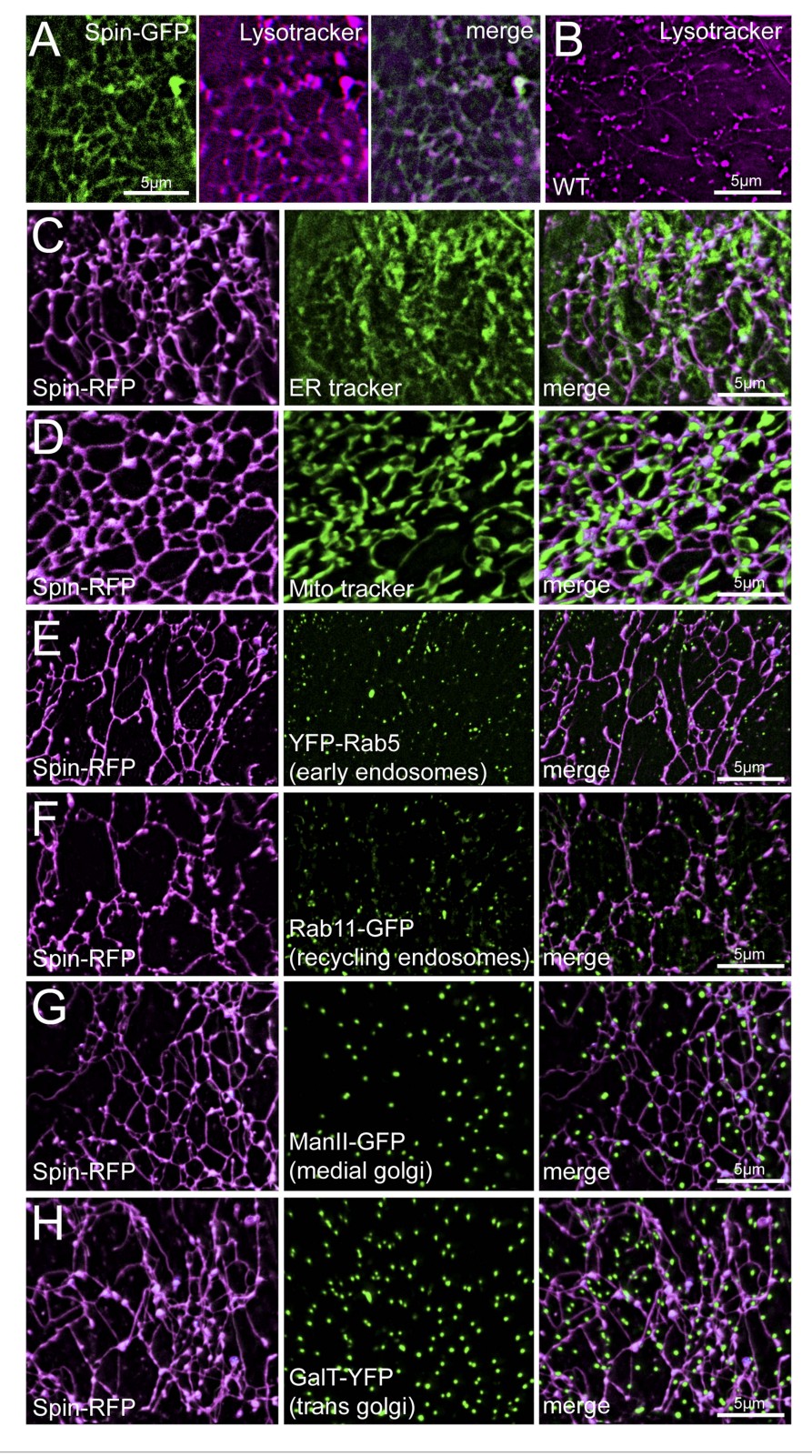

**Figure 2**. Spin-RFP tubules do not co-localize with mitochondria, ER, golgi or early endosomes. (**A**) Co-imaging of Spin-GFP and Lysotracker Red staining. (**B**) Lysotracker staining of wild type muscles. (**C**–**H**) Co-imaging of Spin-RFP with ER tracker (**C**), Mito tracker (**D**), YFP-Rab5 (**E**), Rab11-GFP (**F**), ManII-GFP (**G**), GalT-YFP (**H**).

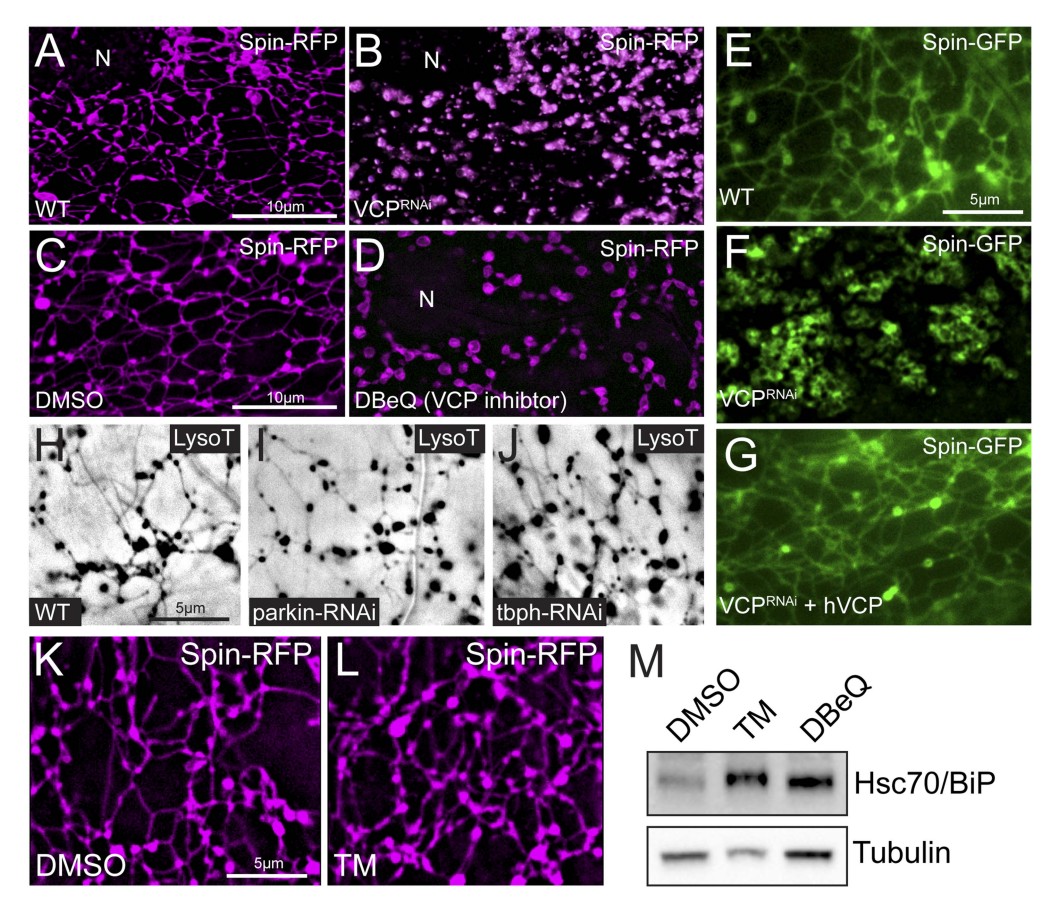

**Figure 3**. VCP inhibition disrupts the lysosome tubule lattice and human VCP rescues this defect. (**A**) Representative live image of *Spin-RFP* expressed in muscle using the muscle-specific *BG57-Gal4* driver. (**B**) Live image of Spin-RFP in muscles expressing *VCP-RNAi* using the muscle-specific *BG57-Gal4* driver. (**C**, **D**) Live images of Spin-RFP expressed in muscles that were treated with DMSO (**C**) or the VCP-specific inhibitor DBeQ (**D**) for 4 hr. (**E**) Live image of *Spin-GFP* expressed in muscles using the muscle-specific *BG57-Gal4* driver. (**F**) Live image of Spin-GFP in muscles expressing *VCP-RNAi* using the muscle-specific *BG57-Gal4* driver. (**G**) Live image of Spin-GFP in muscles that co-express *VCP-RNAi* and human *VCP* (hVCP) using the muscle-specific *BG57-Gal4* driver. (**H–J**) Lysotracker staining in wild type (**H**) muscles or muscles expressing *parkin-RNAi* (**I**) or *tbph-RNAi* (**J**). (**K**, **L**) Spin-RFP localization in muscles treated with DMSO (**K**) or tunicamycin (TM) (**L**). (**M**) Western blot analysis of total Hsc70/BiP protein levels. Tubulin serves as a loading control.

Since VCP knockdown destroys the sarcoplasmic tubular lysosomal network prior to obvious cellular degeneration, we speculated that dismantling of the network could be a precursor to cellular degeneration. However, it is also possible that loss of this network is a secondary correlate of impaired muscle health. To address this issue, we examined two additional RNAi-mediated conditions that cause muscle degeneration, looking for the presence or absence of tubular lysosomes. Specifically, we expressed RNAi against *parkin* and *tbph*, *Drosophila* orthologues of genes linked to Parkinson's and ALS, respectively. These RNAi have been shown to cause muscle degeneration in *Drosophila* (*Diaper et al., 2013*; *Cornelissen et al., 2014*). Remarkably, we did not observe any significant effect on lysosome tubules (*Figure 3H–J*) when *parkin-RNAi* and *tbph-RNAi* are expressed with *BG57-Gal4*, the same Gal4 line used to express *VCP-RNAi* throughout our studies.

VCP has well-established roles in the ERAD pathway and loss of VCP causes ER stress (*Wójcik et al., 2006*). Thus, it is possible that loss of tubular lysosomes is caused indirectly by ER stress. To address this possibility, we treated muscles with tunicamycin (TM) to induce ER stress by another

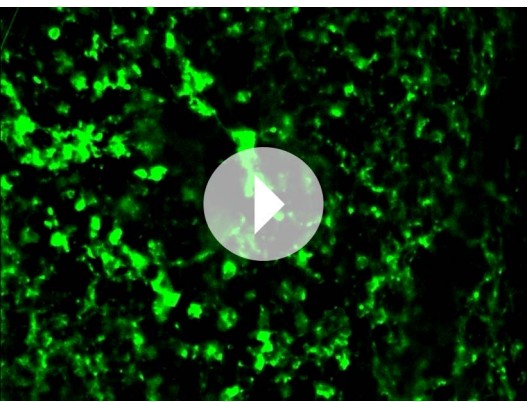

**Video 4.** Spin-GFP dynamics in muscles expressing *VCP-RNAi*. Representative time-lapse video of Spin-GFP in muscles expressing *VCP-RNAi*. Frames were taken at 10 s intervals.

means and examined the tubular network. Upon treatment with TM for 4 hr, the Spin-RFP tubular network remained intact (*Figure 3K,L*). We verified that ER stress was induced by examining levels of Hsc70/BiP, a marker of ER stress, in protein lysates derived from the same animals that were used for imaging. Total protein levels of Hsc70/BiP were significantly increased in both TM and DBeQ treated muscles (*Figure 3M*). Thus, tubular network disruption is not a byproduct of ER stress. Taken together, our data are consistent with three conclusions: (1) VCP loss of function disrupts the integrity of the tubular lysosomal network, (2) the role of VCP in maintaining tubular lysosomes is conserved and (3) disruption of the tubular network is specific for VCP loss of function and not a secondary byproduct of muscle degeneration or ER stress.

## Autophagosome membranes co-localize with the tubular lysosome network

Muscle biopsies from patients with VCP-related disease display an accumulation of cytoplasmic poly-ubiquitin aggregates (*Watts et al., 2004*; *Weihl et al., 2009*; *Dolan et al., 2011*), suggesting a defect in protein clearance. This led us to explore the intersection of the observed tubular lysosomal network and autophagy. First, we co-imaged Spin-GFP with the autophagosome membrane marker mCherry-Atg8a/LC3 to determine the relationship between autophagosomes and the tubular lysosome network. Remarkably, we find that mCherry-Atg8a precisely co-localizes with Spin-GFP-labeled tubules (*Figure 4A*). The mCherry-Atg8a labeling is widely distributed within the network, but does not label the full extent of every tubule (*Figure 4A*). Time-lapse imaging revealed mCherry-Atg8a labeling is dynamic within Spin-GFP tubules (*Video 5*). Some mCherry-Atg8a puncta appear to traffic through the Spin-GFP tubules suggesting that autophagosome membranes and/or cargos are dynamic within the tubular lysosome network (*Figure 4B*). Atg8 localizes to the autophagophore membrane prior to the formation of an enclosed autophagosome, at which point Atg8 is shed from the outer autophagosome membrane (*Xie and Klionsky, 2007*). After autophagosomes fuse with lysosomes, lysosomal enzymes degrade the inner autophagosome membrane containing Atg8 (*Xie and Klionsky, 2007*). Given this sequence of events, we propose that the mCherry-Atg8a positive regions of the lysosome tubules represent ongoing degradation of autophagosome membranes following fusion with the Spin-GFP positive lysosomal network.

Next, we explored the consequence of disrupting VCP activity on Atg8 and Spin-GFP co-localization. When VCP was knocked down and the tubular-lysosomal network was eliminated, Atg8 no longer co-localized with Spin-GFP (*Figure 4C*). Rather, Atg8-positive vesicles were found closely apposed to Spin-positive, vesicular lysosomal compartments. This close apposition suggests that the autophagosomes can identify and potentially dock against the lysosomal membranes, but fusion of the autophagosome with the lysosome is defective. This is consistent with an established role for the yeast VCP homolog, Cdc48, in membrane fusion (*Latterich et al., 1995*).

To test whether tubules that are positive for both mCherry-Atg8a and Spin-GFP are, indeed, a consequence of fused auto-lysosomes, we imaged a dual fluorescent reporter GFP-mCherry-Atg8a (*Nezis et al., 2010*). In neutral pH conditions, both GFP and mCherry fluoresce. However, in acidic environments such as in the lysosomal lumen, GFP fluorescence is quenched and only mCherry is observed. Thus, autophagosomes that are fused with lysosomes will exhibit mCherry but not GFP fluorescence. It is important to note that Spin-GFP fluorescence is not quenched because the C-terminal GFP tag resides on the cytoplasmic face of lysosomes (*Dermaut et al., 2005*). When we expressed the dual fluorescence reporter in muscles we found no detectable GFP fluorescence in the tubules that label strongly for mCherry-Atg8a (*Figure 4D*), consistent with other data indicating that

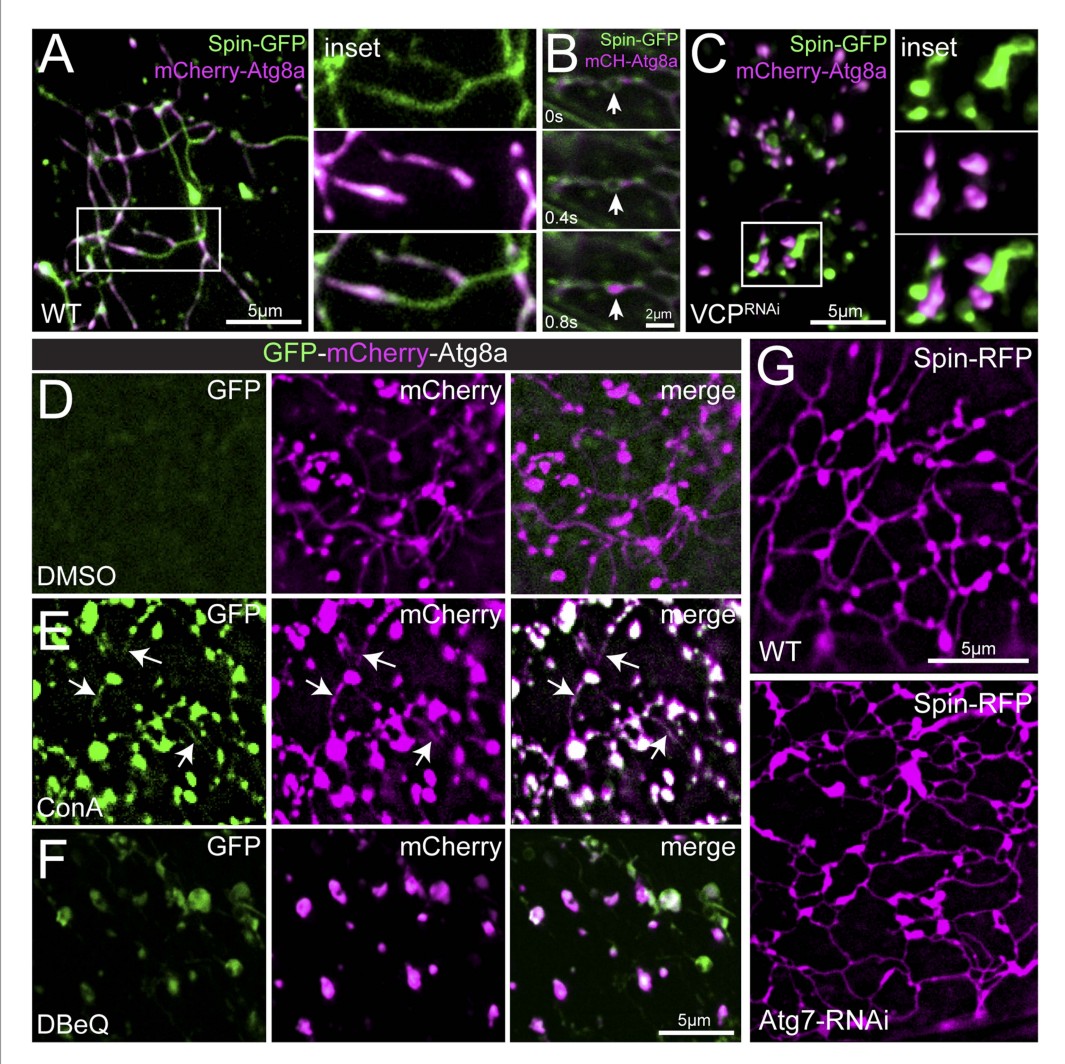

**Figure 4**. Autophagosomes co-localize with the tubular lysosomal network. (**A**) Representative live image of Spin-GFP and mCherry-Atg8a co-expressed in muscles using the muscle-specific *BG57-Gal4* driver. (**B**) Representative time-lapse sequence of Spin-GFP and mCherry-Atg8a in muscle. The arrow follows a mCherry-Atg8a positive puncta trafficking along a Spin-GFP tubule. (**C**) Spin-GFP and mCherry-Atg8a no longer co-localize in muscles expressing *VCP-RNAi*. White box indicates region shown at higher magnification and separate channels at right. **D**. Live image of GFP-mCherry-Atg8a in muscles treated with DMSO for 3 hr. Note the lack of GFP signal. (**E**) Live image of GFP-mCherry-Atg8a in muscles treated with the V-ATPase specific inhibitor Concanamycin A (ConA) for 3 hr. Note the presence of GFP-positive tubules. (**F**) Live image of GFP-mCherry-Atg8a in muscles treated with the VCP-specific inhibitor DBeQ for 3 hr. Note the presence of GFP-positive vesicles. (**G**) Spin-RFP localization in WT muscles or muscles expressing Atg7-RNAi using the muscle specific *BG57-Gal4* driver.

the tubules are acidic. To further test the acidic nature of the tubules, we treated muscles with concanamycin A, a specific inhibitor of the lysosomal V-ATPase that is required for lysosome acidification (*Huss et al., 2002*). Upon treatment with ConA for 3 hr, we observed GFP and mCherry positive tubules (*Figure 4E*). Together, these data further verify that the lysosome tubules are acidic and also demonstrate that acidification is not necessary to maintain the structure of the tubules.

Again, we explored the consequence of disrupting VCP activity. Inhibiting VCP function with DBeQ created GFP-positive vesicles (*Figure 4F*) that must be non-acidified compartments, a finding that is consistent with the existence of autophagosomes that have not yet fused with Spin-positive lysosomes. Since the appearance of GFP-positive vesicles occurs in a time frame of minutes to hours, it

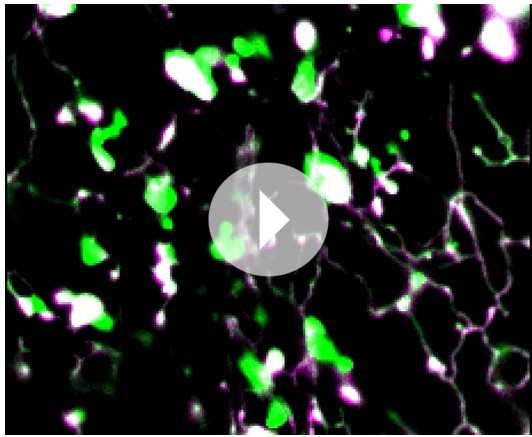

**Video 5.** Spin-GFP and mCherry-Atg8a dynamics in *Drosophila* muscle. Representative time-lapse video of Spin-GFP and mCherry-Atg8 co-expressed in muscles. Frames were taken at 10 s intervals.

suggests that there is a continual flux of material through the auto-lysosomal system in muscle cells at steady state.

Finally, we examined whether autophagy influx was required to induce or maintain lysosome tubules. Expression of *Atg7-RNAi*, an RNAi line that has been shown previously to inhibit autophagosome assembly (*Ren et al., 2009*), did not affect lysosome tubules (*Figure 4G*). Thus, autophagy induction is not a prerequisite for lysosome tubules. Taken together, these data confirm previous reports that VCP is required for autophagosome-lysosome fusion (*Tresse et al., 2010b*) and further suggest that VCP loss abrogates autophagosome-lysosome fusion by affecting the structural properties of lysosomes.

## VCP localizes to auto-lysosomes

Previous studies have reported that, when overexpressed in cultured cells, mammalian VCP localizes diffusely at the nucleus and throughout the cytoplasm (*Vesa et al., 2009*; *Tresse et al., 2010a*; *Wang et al., 2011*). To examine VCP localization in live muscles, we generated a *UAS-VCP-Venus* transgenic fly. Similar to previous reports for mammalian VCP, we observed abundant VCP localization in and around the nucleus and diffusely in the cytoplasm (*Figure 5—figure supplement 1*). But, VCP-Venus also concentrated at structures that are labeled by either Spin-RFP (*Figure 5A*) or mCherry-Atg8a (*Figure 5B*), demonstrating that VCP localizes to auto-lysosomes.

Because inhibiting the catalytic function of VCP leads to tubule deterioration, we tested whether inhibiting the catalytic function of VCP would affect its localization to auto-lysosomes. Surprisingly, inhibiting VCP activity with DBeQ for 2 hr triggered the formation of rod-shaped VCP aggregates that effectively sequester VCP-Venus from the sarcoplasm (*Figure 5C*). The formation of these aggregates is striking, as they resemble rod-like structures characteristic of prion aggregates. We considered two possible reasons that application of DBeQ might increase the propensity for VCP to aggregate. First, DBeQ binding to VCP might initiate aggregation directly by altering the solubility of VCP. Alternatively, the aggregation could be caused by the functions of VCP in other contexts, such as proteasome-dependent protein degradation. Remarkably, when we applied the proteasome inhibitor MG132, VCP-Venus rapidly aggregated in a manner identical to that observed following DBeQ incubation (*Figure 5D*). VCP aggregation was not due to increased VCP protein levels as a result of proteasome inhibition, because total VCP protein did not increase significantly upon MG132 or DBeQ treatment (*Figure 5E*).

While the significance of VCP-Venus aggregates remains uncertain, this phenotype provided us with a means to rapidly and reversibly sequester VCP protein and control its access to auto-lysosomal membranes. Since VCP exists as a hexamer, MG132 incubation in muscles over-expressing VCP-Venus should sequester both wild type and Venus-tagged VCP. Application of MG132 induced VCP-Venus aggregate formation, which correlated with dissolution of the tubular lysosomal network (*Figure 5D*). When MG132 was washed out, VCP aggregates dissolved within 20 min and, as they disappeared, cytoplasmic VCP fluorescence intensity increased (*Figure 5F* and *Video 6*). During this time, cytoplasmic VCP-Venus accumulated at autophagosomes/lysosomes (mCherry-Atg8a) and tubules began to reform (*Figure 5F* and *Video 6*). These data indicate that VCP localizes to auto-lysosomes, where it could participate in auto-lysosomes tubulation.

## Disrupted lysosome tubules correlate with muscle weakness and autophagy/lysosome defects

We next investigated the consequence of disrupting the auto-lysosome tubule network. We first examined overall muscle function. When VCP RNAi was expressed specifically in muscle, muscle

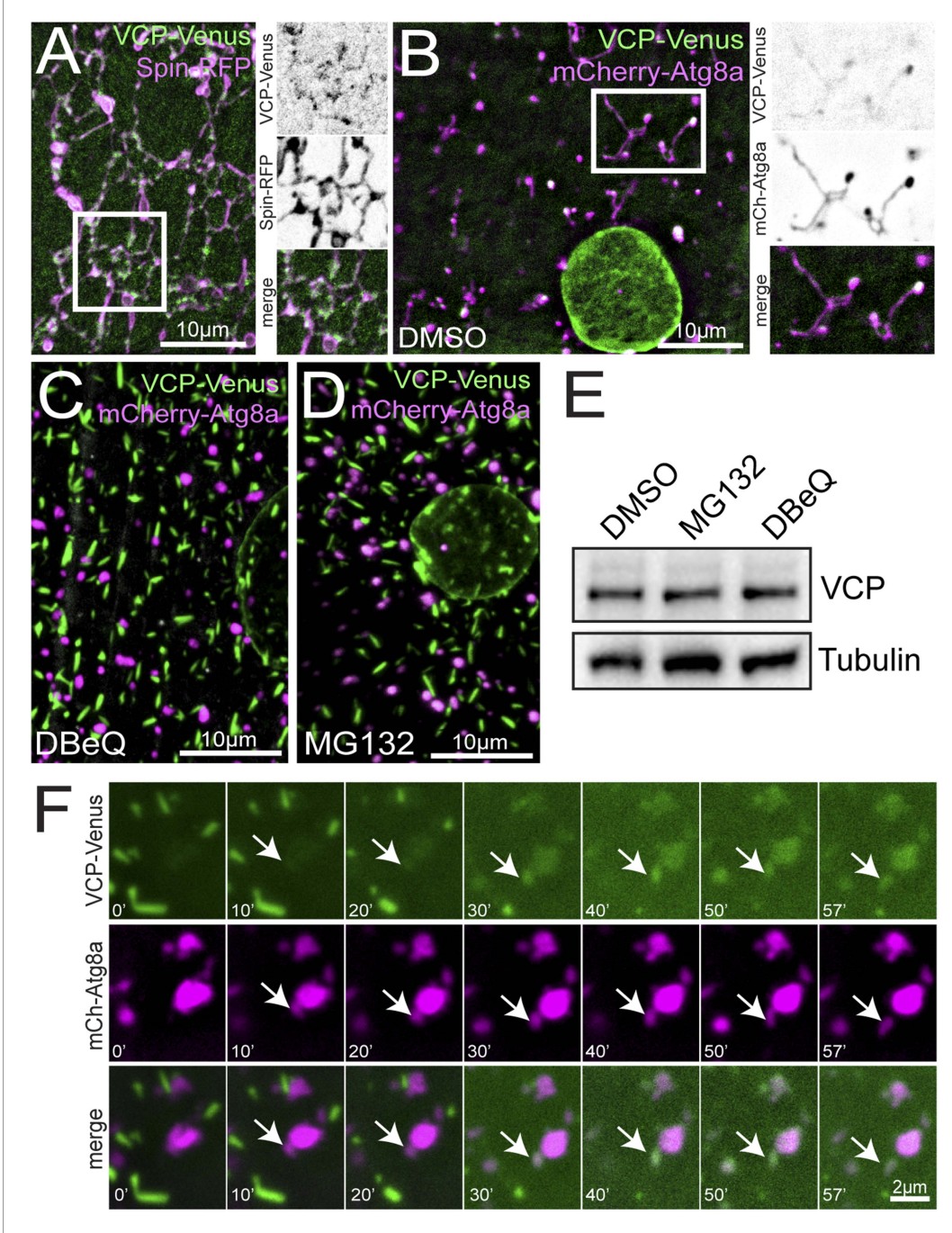

**Figure 5**. VCP co-localizes with the tubular auto-lysosomes. (**A**) Representative live image of VCP-Venus and Spin-RFP expressed in muscles using the muscle-specific *BG57-Gal4* driver. White box indicates region shown at higher magnification and separate channels at right. (**B**) Representative live image of VCP-Venus and mCherry-Atg8a expressed in muscles using the muscle-specific *BG57-Gal4* driver. Inset as in **A**. (**C**, **D**) VCP-Venus and mCherry-Atg8a localization in muscles treated with the VCP inhibitor DBeQ (**C**) or the proteasome inhibitor MG132 (**D**) for 3 hr. (**E**) Western blot analysis of total VCP protein levels from muscles in the treatments indicated. Tubulin serves as a loading control. (**F**) Representative time-lapse sequence of VCP-Venus and mCherry-Atg8a after MG132 was washed out. The arrow indicates a tubule extending from a mCherry-Atg8a positive vesicle.

*Figure 5. continued on next page*

*Figure 5. Continued*

The following figure supplement is available for figure 5:

**Figure supplement 1**. VCP-Venus localization in *Drosophila* muscle.

wasting was apparent and third instar larvae exhibited a severe impairment in their ability to move (*Figure 6—figure supplement 1A,B*). When prodded, the animals would move, but their movements were slow and only lasted for short periods of time. The defect in their motility is not due to defects in the nervous system because synaptic transmission at the NMJ remained intact (*Figure 6—figure supplement 1C,D*). These data are consistent with compromised muscle function that parallels the muscle weakness observed in human VCP-related diseases.

Next, we investigated the degradation capacity of the collapsed tubular lysosomes. For lysosomes to degrade their cargo they must be acidified and the proteolytic enzymes must be present. We first examined the acidification of the lysosomes by co-imaging Spin-GFP with Lysotracker in *VCP-RNAi* animals and found that the enlarged Spin-GFP vesicles also co-stained with Lysotracker (*Figure 6—figure supplement 2A*), indicating that they are acidic. Then, we examined whether lysosome enzymes were delivered properly to the lysosomes. Normally, the lysosomal enzyme Cathepsin-L is proteolytically processed in the lysosomal lumen to form a mature enzyme. To further examine the functionality of the lysosomes, we examined processing of Cathepsin-L and found no significant difference in Cathepsin-L processing in *VCP-RNAi* animals or animals treated with DBeQ (*Figure 6—figure supplement 1B*). Thus, disruption of the tubular lysosomal network does not appear to affect the proteolytic capacity of sarcoplasmic lysosomes.

To this point, our data suggest that loss of VCP disrupts the fusion of autophagosomes with functional lysosomes and, in parallel, causes the collapse of the tubular lysosomal network. Based on this, we expected to find evidence of failed autophagy in VCP knockdown muscle. In wild type muscles stained with a poly-ubiquitin antibody, we observed small puncta around the nucleus and a few small puncta in the cytoplasm (*Figure 6A*). These small puncta likely represent active sites of protein degradation by the proteasome. However, in muscles expressing *VCP-RNAi* we observed a dramatic accumulation of cytoplasmic poly-ubiquitin aggregates (*Figure 6B*). Even acute treatment with DBeQ was sufficient to produce cytoplasmic poly-ubiquitin aggregates (*Figure 6C–E*). We also note that VCP inhibition caused a dramatic decrease in poly-ubiquitin conjugates around the nucleus, which likely reflects failed delivery of poly-ubiquitinated proteins to the proteasome. Importantly, we find that Spin-positive lysosomes are devoid of poly-ubiquitin staining (*Figure 6F*), an observation that is consistent with our model of failed fusion of autophagosomes with lysosomes.

In addition to clearing protein aggregates from the cytoplasm, autophagy is also responsible for clearing damaged organelles, including mitochondria. We examined the functional pool of muscle mitochondria in vivo using mitotracker-Orange CM-H2TMRos, which selectively stains mitochondria with an active membrane potential. Muscles expressing *VCP-RNAi* displayed less mitotracker-Orange CM-H2TMRos staining and the visualized mitochondria had an altered morphology, appearing round and dispersed rather than being densely packed, tubular structures (*Figure 6G,H*). The swollen mitochondria observed in the VCP RNAi expressing muscles

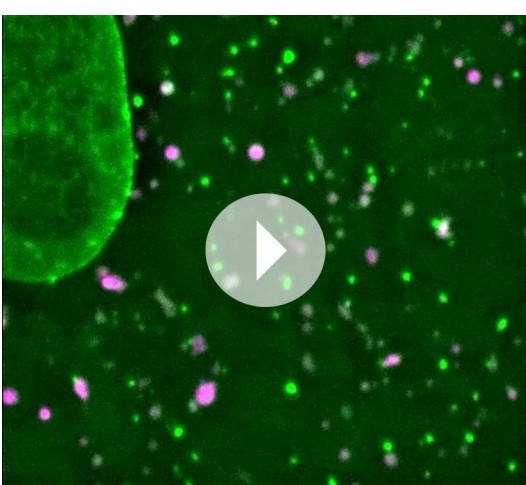

**Video 6.** VCP-Venus and mCherry-Atg8a dynamics after MG132 wash out. Muscles co-expressing VCP-Venus and mCherry-Atg8 were treated with the proteasome inhibitor MG132 for 3 hr. MG132 was washed out and time-lapse images were taken every 10 s.

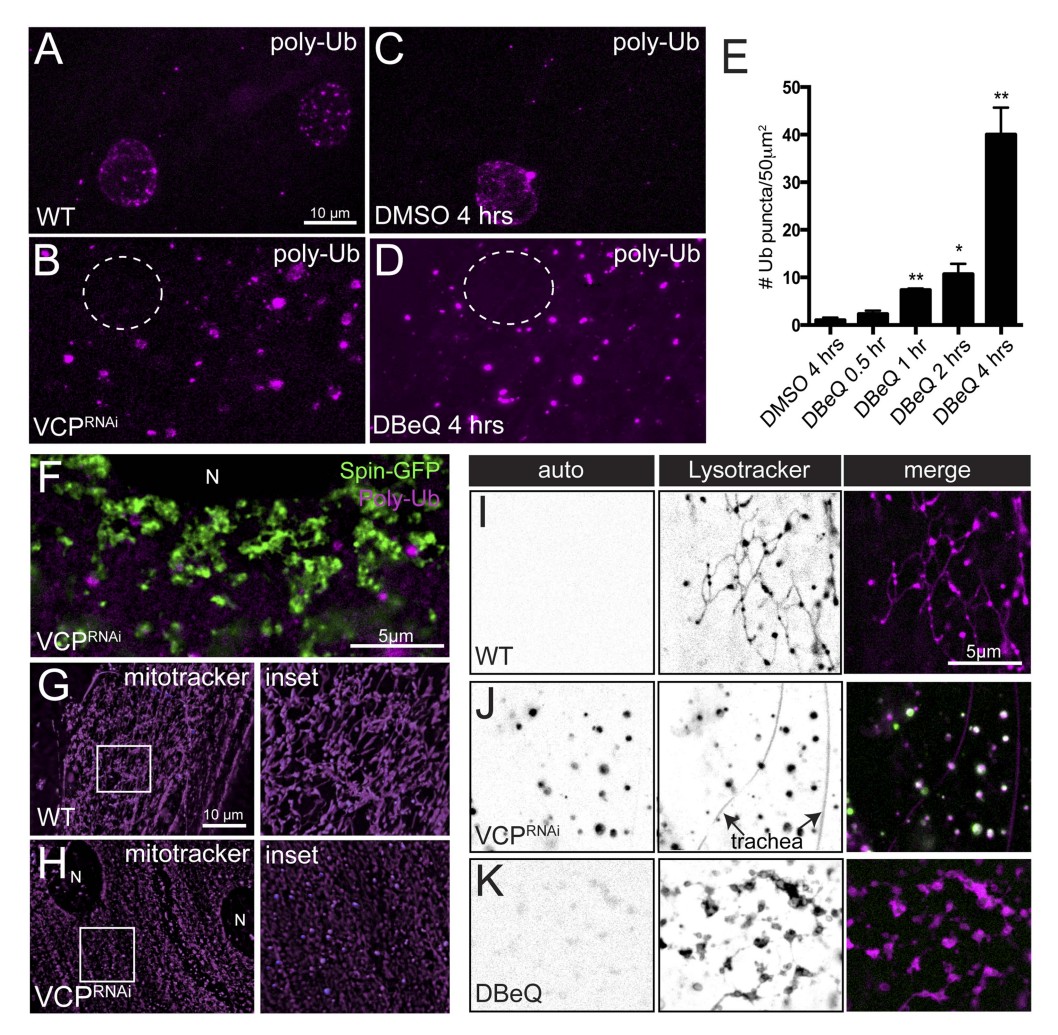

**Figure 6**. Disruption of the tubular auto-lysosomal network correlates with increased poly-Ubiquitin aggregates, impaired mitochondria and increased lipofuscin granules. (**A**, **B**) Wild type (**A**) and *VCP-RNAi* (**B**) expressing muscles were fixed and stained with a poly-Ubiquitin antibody. Nuclei with localized poly-Ubiquitin staining are apparent in **A**. Nuclei are indicated (dashed circle) in **B**. (**C**, **D**) Wild type animals were treated with DMSO (**C**) or the VCP-specific inhibitor DBeQ (**D**), fixed and stained with a poly-Ubiquitin antibody. (**E**) Quantitation of the number of poly-Ubiquitin aggregates per 50 $\mu m^2$ from wild type muscles treated with DMSO for 4 hr or DBeQ for various times (n = 9, *p < 0.05, **p < 0.01). (**F**) Localization of Spin-GFP and poly-Ubiquitin in muscles expressing *VCP-RNAi*. (**G**, **H**) Mitotracker-C2TMRos staining in control (**G**) and *VCP*-RNAi (**H**) muscles. (**I–K**) Autofluoresence at 488 nm and lysotracker staining in wild type (**I**), muscles expressing *VCP-RNAi* (**J**), and wild type muscles treated with the VCP-specific inhibitor DBeQ for 4 hr (**K**).

The following figure supplements are available for figure 6:

**Figure supplement 1**. Loss of tubular lysosomes correlates with impaired muscle function.

**Figure supplement 2**. Lysosomal acidity and Cathepsin processing are maintained in *VCP-RNAi* expressing muscles.

are likely defective and should be a prime target for mitophagy-dependent degradation. Taken together with the appearance of polyubiquitin aggregates, these data are consistent with an overall defect in autophagic clearance of proteins and defective organelles. However, since VCP has also

been shown to be recruited directly to damaged mitochondria (*Kim et al., 2013*) we cannot rule out the possibility that the effects on mitochondrial morphology and membrane potential are due to direct VCP functions at the mitochondrial outer membrane.

Finally, we noted an accumulation of lipofuscin granules in *VCP-RNAi* expressing muscles. Lipofuscin granules are a conglomerate of polymerized non-degradable proteins and lipids that build up in the lysosomal lumen (*Szweda et al., 2003*). A distinguishing property of lipofuscin granules is that they exhibit auto-fluorescence at 488 nm. Wild type or *VCP-RNAi* muscles were stained with LysoTracker-Red and imaged at both 488 nm (green) and 555 nm (magenta). The wild type muscles did not display any detectable auto-fluorescence at 488 nm (*Figure 6I*). But, we observed strong auto-fluorescent puncta in *VCP-RNAi* that co-localized with LysoTracker (*Figure 6J*), indicative of lipofuscin granules. We did not observe autofluorescent puncta to the same extent when wild type muscles were treated with DBeQ for 4 hr (*Figure 6K*), suggesting that lipofuscin accumulation is a progressive phenotype. Alternatively, lipofuscin granule accumulation could require loss of VCP protein rather than just loss of VCP catalytic activity. These data suggest that maintaining the structural integrity of lysosome tubules is critical for lysosome function.

## VCP-related disease mutations disrupt lysosome tubules

Finally, we asked whether overexpression of *VCP* transgenes that harbor disease-causing mutations impairs the presence or dynamics of tubular lysosomes. To date, a total of 19 missense mutations in 13 different residues are associated with IBMFD that reside in either the Cdc48 homology domain, the L1 linker domain or the D1 ATPase domain (*Figure 7A* and [*Ju and Weihl, 2010*]). We selected one mutation from each of these three domains to examine the effect on lysosome tubules: R155H, R191Q and A232E. R155 is the most common hereditary mutation and is located in the CDC48 homology domain, which is a protein interaction module that plays an important role in VCP substrate binding (*Ju and Weihl, 2010*). R191 is located in the linker region between the CDC48 domain and the first ATPase catalytic domain (D1). A232 is located at the beginning of the of the D1 domain and is the most severe clinical mutation (*Watts et al., 2004*). All three residues are conserved in the *Drosophila* protein (*Figure 7A*). As a control, we show that over-expression of wild type *dVCP* does not alter tubular lysosomes (*Figure 7B*). We then demonstrate that over-expression of each of the mutant *VCP* transgenes profoundly impairs the tubular lysosomal network (*Figure 7C–F*). In parallel, we find an increase in sarcoplasmic auto-fluorescence when these *VCP* mutants are over-expressed (*Figure 7B–G*). Overexpression of the A229E mutation, which is the most severe clinical mutation, caused the largest increase in auto-fluorescence compared to the other clinical mutations (*Figure 7F*). Because over-expression of the disease relevant VCP transgenes phenocopies *VCP-RNAi* expression, these data suggest that the disease mutations are dominant interfering mutations. Thus, over-expression of *VCP* disease-associated mutations disrupts lysosome tubules in vivo, an effect that causes accumulation of cytoplasmic lipofuscin granules.

## Discussion

Here we demonstrate that lysosomes form a dynamic, tubular array that extends throughout the sarcoplasm of *Drosophila* muscle, in vivo. To our knowledge, this is the first observation of such an extensive, dynamic tubular lysosomal network in any in vivo system. We define this as a lysosomal network because it has a low pH, it is labeled by the late endosomal marker Spin-GFP, does not co-localize with the ER, golgi apparatus, early endosomes or recycling endosomes, and we find that mCherry-Atg8 traffics to this compartment, indicative of autophagosome fusion with a network of tubular lysosomes. Lysosomal network dynamics require an intact microtubule cytoskeleton, clathrin heavy chain, and VCP. We provide evidence that the catalytic activity of VCP is continuously required to sustain network integrity. When VCP is depleted or inhibited, the lysosomal network collapses and both cellular proteostasis and autophagy are compromised. Collapse of the network correlates with cellular manifestations of VCP-associated degenerative diseases, including the appearance of protein inclusion bodies (*Kimonis et al., 2008*; *Weihl et al., 2009*). While many questions remain unanswered, our discovery of tubular lysosome dysfunction following overexpression of pathogenic VCP mutants represents one the earliest markers of VCP-dependent muscle pathology. Further characterization could pave the way toward methods that might beneficially stabilize lysosomal function and ameliorate the progression of widespread degenerative disease.

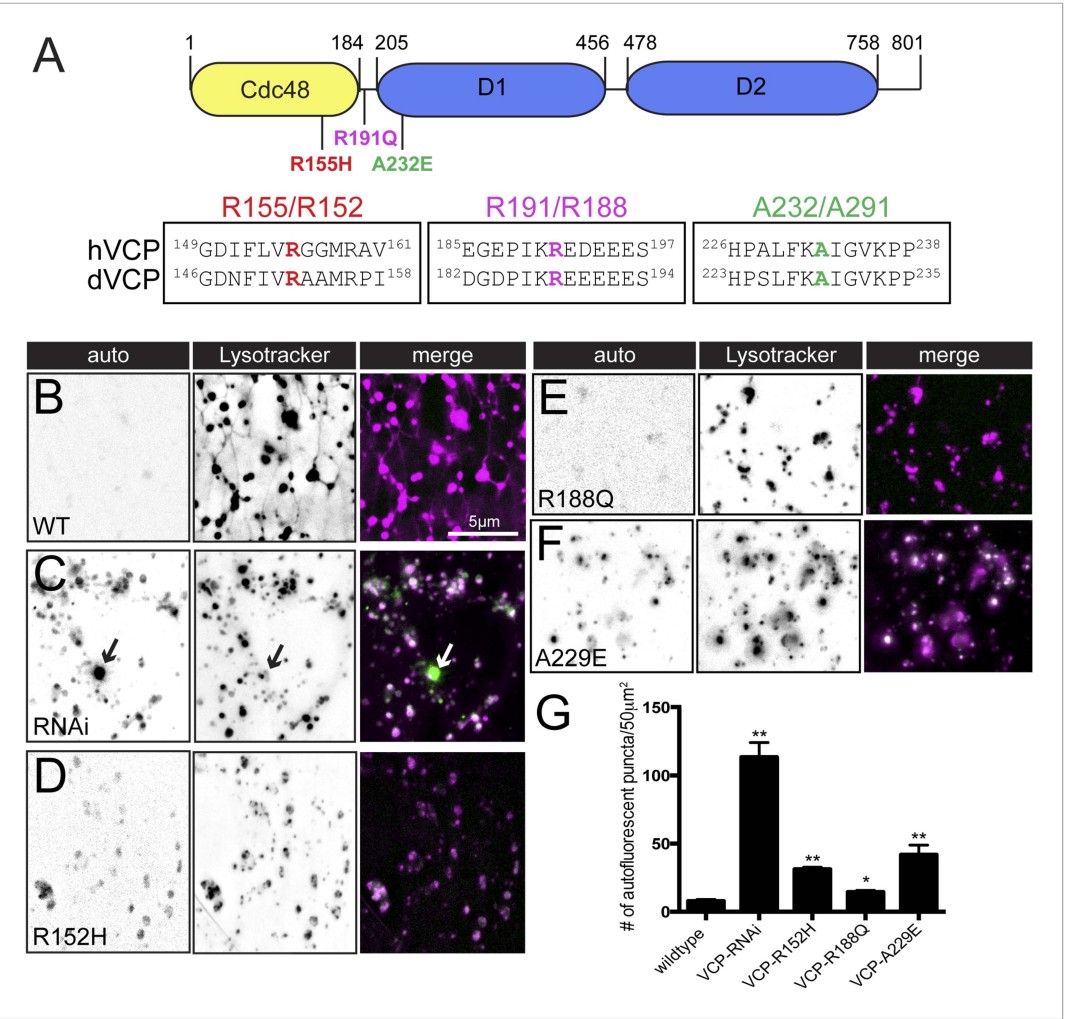

**Figure 7**. Pathogenic VCP alleles disrupt the tubular auto-lysosomal network. (**A**) Schematic diagram of VCP protein. Top: Human pathogenic *VCP* mutations are labeled on the cartoon. Bottom: sequence alignment of human *VCP* and *Drosophila VCP/Ter94* pathogenic mutant regions. (**B–F**) Autofluoresence at 488 nm and lysotracker staining in wild type muscles expressing *VCP-WT* (**B**), *VCP-RNAi* (**C**) *VCP-R152H* (**D**), *VCP-R188Q* (**E**) or *VCPA-229E* (**F**) transgenes. (**G**) Quantitation of the number of auto-fluorescent puncta per 50 µm² in the genotypes indicated (n = 9, *p < 0.05, **p < 0.01).

## Role of an extended tubular lysosome network in muscles

In most tissues, autophagy is induced upon nutrient starvation, but muscles are one of the unique tissues in which autophagy occurs in the absence of starvation (*Mizushima et al., 2004*). In fact, basal autophagy is required to maintain muscle mass (*Masiero et al., 2009*). The constitutive levels of autophagy in muscle are likely related to the large energy requirement of muscles compared to other cell types and their ability to serve as a source of metabolic energy for other organs (*Sandri, 2010*). Muscle sarcoplasms are also unique in that they are shared between multiple nuclei and are much larger in volume compared to many other cell types. We propose that the observed extended, lysosomal network that is maintained by VCP is a cellular solution to ensure highly efficient autophagy throughout the entirety of muscle sarcoplasms. It is as if the lysosomes vascularize the sarcoplasm, ensuring that no portion of the sarcoplasm is ever far from a lysosomal depot where autophagic cargo can be degraded. Likewise, this network could facilitate the local recycling of nutrients throughout the sarcoplasm to effectively meet local demands. A similar argument can be made regarding the autophagy-dependent turnover of mitochondria, which are densely distributed throughout the

muscle. In support of this, VCP is critical for the rapid degradation of muscle proteins during muscle atrophy and expression of a dominant negative form of VCP reduces protein degradation by both proteasomes and lysosomes (*Piccirillo and Goldberg, 2012*).

Lysosomal membranes have been observed to extend tubules and fission off to produce de novo lysosomes following fusion of autophagosomes with lysosomes (*Yu et al., 2010*). This process, termed ALR, serves as a mechanism to recycle lysosomes when autophagic demand is high (*Yu et al., 2010*). It is unknown whether ALR is a constitutive process that participates in ongoing, steady-state proteostasis in vivo or whether it is a process that is specifically induced following stress-induced autophagy. It is possible that the extended, dynamic lysosomal tubule network that we observe is related to ALR, perhaps acting as a platform for efficient ALR throughout the sarcoplasm. In support of this, the mammalian *Spinster* homolog is required for ALR (*Rong et al., 2011*). However, the relatively low frequency with which we observed scission events and the formation of new small lysosome vesicles is not entirely consistent with this idea.

The tubular network identified in this study more closely resembles the lysosome tubular networks that have been observed in cultured macrophage cells (*Swanson et al., 1987a*, *1987b*; *Knapp and Swanson, 1990*). Macrophage lysosome tubules can form a web of tubules that appear to be stably connected throughout the cytoplasm (*Swanson et al., 1987b*). The purpose of lysosome tubules in this system is still mysterious. We have now identified specific defects associated with loss of lysosome tubules and our data suggest that the tubular network may distribute auto-lysosomal activity throughout the cell. Clues to the function of tubular lysosomes might also be gleaned from studies of early endosomes. Early endosomes form vesicular-tubular structures similar to what we have observed here (*Huotari and Helenius, 2011*). In early endosomes it has been established that tubules can be discrete membrane compartments with a different lipid composition and cargo compared to that within the vesicular body of the early endosome (*Huotari and Helenius, 2011*). By analogy, the ability of lysosomes to exist in a tubular-vesicular state in cellular contexts where their functional demand is high, might allow lysosomes to execute diverse functions more efficiently through compartmentalization.

## Consequences of tubular lysosome dysfunction

VCP and the yeast homologue Cdc48 have been ascribed many functions within the cell including cell cycle progression (*Moir et al., 1982*), UPS and ERAD protein degradation (*Meyer et al., 2012*; *Wolf and Stolz, 2012*), mitophagy (*Taylor and Rutter, 2011*) and classical autophagy (*Ju et al., 2009*; *Tresse et al., 2010a*; *Dargemont and Ossareh-Nazari, 2012*; *Meyer et al., 2012*). VCP achieves these diverse activities, in part, through its generalized function as an ubiquitin-dependent 'segregase' that dissociates protein conjugates tagged with ubiquitin from protein complexes and organelle membranes (*Halawani and Latterich, 2006*; *Meyer et al., 2012*). For each process that relies upon VCP activity, different cofactors control VCP localization and function (*Baek et al., 2013*). To this list of VCP-mediated activities, we now add the action of VCP in controlling the integrity and dynamics of a tubular lysosomal system and fusion of autophagosomes with tubular lysosomes. Additional, future experimentation will be necessary to determine which specific VCP-mediated molecular mechanism(s) is most directly relevant to the integrity and dynamics of lysosomal tubules in muscle.

The phenotypic consequences following the loss or inhibition of VCP in *Drosophila* muscle include the collapse of the tubular lysosomal network, failed fusion of autophagosomes with lysosomes, accumulation of sarcoplasmic poly-ubiquitin aggregates, accumulation of lipofuscin granules, impaired mitochondria and impaired muscle health. All of these phenotypes are hallmarks of degenerative diseases that are associated with mutations in VCP (*Watts et al., 2004*). We can now ascribe some of these phenotypes to the action of VCP at lysosomal membranes. Our data indicate that VCP localizes to autolysosomes and loss of VCP causes collapse of the tubular lysosomal network and failed autophagosome-lysosome fusion. Furthermore, when previously sequestered VCP is released back into the cytoplasm, VCP translocates to dormant autolysosomes and tubulation ensues. Although we cannot distinguish the role of VCP in the initiation and/or maintenance of lysosome tubules, these data argue that VCP acts at the autolysosmal membrane to control autolysosomal dynamics and the progression of autophagic protein clearance. If VCP is necessary for normal activity of the autophagy-lysosome system in muscle as our data suggests, then it seems likely that the accumulation of poly-ubiquitin aggregates and lipofuscin granules are a direct consequence of

impaired VCP-dependent lysosomal function. This assertion is further supported by our demonstration that over-expression of pathogenic *VCP* mutant transgenes disrupt the tubular lysosomal network and also cause accumulation of ubiquitin and lipofuscin material. Importantly, these transgenes in *Drosophila* do not appear to disrupt VCP roles in the UPS or ERAD pathways, emphasizing the role of impaired autophagy in pathogenic VCP phenotypes (*Tresse et al., 2010b*; *Chang et al., 2011*). We acknowledge that lipofuscin granule accumulation following overexpression of pathogenic *VCP* mutant transgenes could represent a byproduct of proteotoxic stress caused by defects in the UPS or autophagy/lysosome pathways that eventually lead to the transformation of proteins into non-degradable products. Ultimately, more detailed biochemical analyses will be required to elucidate precisely how VCP functions at lysosome membranes and how this activity might be coordinated with other aspects of VCP function throughout the cell. Taken together, our identification of tubular lysosomes in *Drosophila* muscle challenges the traditional view of vesicular lysosomes and suggests that lysosome structures can be more versatile than previously assumed. Understanding how lysosomes regulate their morphological state will be an exciting avenue for future studies.

## Materials and methods

### Experimental procedures

#### Fly stocks

The following transgenic fly stocks were generated in this study: *UAS-Spin-RFP*, *UAS-Spin-GFP* and *UAS-Ter94/VCP-Venus*. *Drosophila spinster* and *ter94* cDNA were obtained by amplifying from DGRC (Drosophlia Genomics Resource Center, Bloomington, IN) clones AT25382 and GM02885 respectively. The cDNAs were cloned into the Gateway pENTR vector (Invitrogen, South San Francisco, CA) and subsequently cloned into destination vector pTWV obtained from the *Drosophila* Gateway Vector Collection (Carnegie Institution, Baltimore, MD). Transgenic animals were then generated by BestGene. *UAS-mCherry-Atg8a* and *UAS-GFP-mCherry-Atg8a* stocks were purchased from the Bloomington Stock Center. *UAS-RNAi* stocks were purchased from the Vienna *Drosophila* Resource Center. Pathogeneic *UAS-Ter94* alleles were a kind gift from TK Sang (National Tsing Hua University, Hsinchu, Taiwan). Golgi markers *UAS-ManII-GFP* and *UAS-GalT-YFP* were a kind gift from B Ye (University of Michigan, Ann Arbor, MI). Gal4 muscle drivers used in this study include: *BG57-Gal4* and *MHC-Gal4*.

### Microscopy methods

Third instar larvae were dissected in Schneider's insect cell media (Gibco, Grand Island, NY) supplemented with 10% FBS (Gibco) and Penicillin/Streptomycin (Gibco) and all live imaging was performed in insect cell media. For all imaging experiment, at least 3 muscles in 3 animals were imaged to account for variances between muscles and animals and the most representative images are shown in each figure (n ≥ 9). The following drugs were diluted in insect cell media to the following final concentrations: 1uM LysoTracker Red DND-99 (Life Technologies, South San Francisco, CA), 1uM ER Tracker green (Life Technologies), 10uM Nocodazole (Sigma, St. Louis, MO), 10uM Latrunculin A (Life Technologies), 10uM DBeQ (Sigma), 1uM MG132 (Sigma), 100 nM Concanamycin A (Santa Cruz Biotechnology, Dallas, TX), 2ug/ml TM (Sigma) and 500 nM Mitotracker-Orange-CM-H2TMRos (Life Technologies). LysoTracker, ER Tracker and Mitotracker-Orange-CM-H2TMRos were incubated on the dissected larvae for 1 hr prior to imaging. Nocodazole, LatA, DBeQ, TM, Concanamycin A and MG132 were incubated on the dissected larvae for 3–4 hr prior to imaging.

For Ubiquitin staining, larvae were dissected and fixed with 4% PFA for 15 min. After fixation, larvae were washed 4× with PBS-T and incubated with anti-poly-Ubiquitin (Thermo Scientific, Waltham, MA) at 1:1000 dilution overnight at 4˚C. Larvae were washed again 4× with PBS-T, incubated with a fluorescently labeled secondary antibody at 1:5000 dilution (Life Technologies), and washed again 4× with PBS-T before mounting on a slide with vectashield for imaging.

Imaging was performed on an inverted Axiovert 200 microscope (Zeiss) using a 100× Plan Apochromat objective (1.4NA). Images were captured with a CoolSnap HQ2 CCD camera (Photometrics) and de-convolved using Slidebook 5.0 software (Intelligent imaging innovations, Denver, CO). Image quantification was performed with imageJ software (NIH). Volume rendering was performed with Slidebook 5.0 software. Any adjustment of brightness or contrast was performed using Slidebook 5.0 software, and always applied to the entire image.

## Western blotting methods

Third instar larvae were dissected and muscle preparations were immediately transferred into 5× SDS sample buffer and denatured by boiling for 10 min. Proteins were resolved by SDS-PAGE on a 4–12% Bis-Tris gel (Life Technologies), transferred to a nitrocellulose membrane, immunoblotted with primary and HRP-conjugated secondary antibodies (Life Technologies) and detected using an ECL chemi-luminescent reagent (Life Technologies). The following primary antibodies were used: anti-VCP (Cell Signaling, Danvers, MA) at 1:1000, anti- GRP78/HSPA5 (Thermo Scientific) at 1:2500, insect anti-Cathepsin L (R&D Systems, Minneapolis, MN) at 1:1000, and anti-Tubulin E7-c (Developmental Studies Hybridoma Bank, University of Iowa, IA) at 1:10,000. Secondary HRP conjugated antibodies (GE Lifesciences, Pittsburgh, PA) were used at 1:5000.

## Electrophysiology methods

Sharp-electrode recordings were made from muscle 6 in abdominal segments 2 and 3 from third-instar larvae using an Axoclamp 900A amplifier (Molecular Devices), as described previously (*Frank et al., 2006*). Recordings were made in HL3 saline containing the following components: NaCl (70 mM), KCl (5 mM), MgCl2 (10 mM), NaHCO3 (10 mM), sucrose (115 mM), trehalose (5 mM), HEPES (5 mM), and CaCl2 (0.3 mM). Mean EPSP, mEPSP amplitude, and Rin were obtained by averaging values across all NMJs for a given genotype. EPSP traces were analyzed with custom-written routines in MATLAB (Mathworks, Natick, MA) as previously described (*Gaviño et al., 2015*). mEPSP traces were analyzed in IGOR Pro 6.3 (Wave-Metrics; custom script submitted with this manuscript).

## Acknowledgements

We thank members of the Davis lab for helpful advice and critically reading the manuscript. We are grateful to Kevin Ford and Ryan Jones for providing custom scripts for EPSP and mEPSP analyses. We also thank TK Sang and B Ye for generously sharing fly stocks. AEJ is a fellow of the Jane Coffin Childs Memorial Fund for Medical Research. This study was supported by NIH grant NS047342 to GWD.

## Additional information

### Competing interests

GWD: Reviewing editor, *eLife.* The other authors declare that no competing interests exist.

### Funding

| Funder | Grant reference | Author |
|---|---|---|
| National Institutes of Health | R37NS047342 | Graeme W Davis |
| Jane Coffin Childs Memorial Fund for Medical Research | Postdoctoral Fellowship | Alyssa E Johnson |

The funders had no role in study design, data collection and interpretation, or the decision to submit the work for publication.

### Author contributions

AEJ, Conception and design, Acquisition of data, Analysis and interpretation of data, Drafting or revising the article; HS, Conception and design, Acquisition of data, Analysis and interpretation of data; AGH, Acquisition of data, Analysis and interpretation of data; AT, Contributed unpublished essential data or reagents; GWD, Conception and design, Drafting or revising the article

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
