## [Decision Letter]

Thank you for sending your work entitled “VCP-Dependent Muscle Degeneration is Linked to Defects in a Dynamic Tubular Lysosomal Network in vivo” for consideration at *eLife*. Your article has been favorably evaluated by K VijayRaghavan (Senior editor and Reviewing editor) and three reviewers.

In the current manuscript, Johnston et al. identify a tubular network in the muscle of *Drosophila*, which appears to arise from the fusion of multiple lysosomal vesicles. This remarkable architecture of the lysosomal system requires the AAA-ATPase Valosyl Containing Protein (VCP), mutations of which are implicated in human degenerative disease. Deletion of VCP (or its replacement with disease-relevant mutants) results in disassembly of the lysosomal tubular network and in a series of defects, including defective autophagy and poly-ubiquitin aggregation, linking this unique architecture of the lysosomal system to maintenance of cellular quality control.

Overview of referee comments:

Based on the overall evaluation and comments arising from the collective reviews, we first outline the broad areas of concern, which can be binned into two key points.

A) The referees agree that the novelty that makes this manuscript potentially suitable for *eLife* is the tubular organisation of the lysosomal network in muscle. A more thorough characterisation of this striking sub-cellular structure will therefore be valuable and necessary. Better quantification and explanation of sample sizes as well as controls are needed to demonstrate that the observation of the lysosomal phenotypes seen are not biased by choice of probes used (Spinster vs. Lysotracker for example). These will much strengthen the characterisation of the cellular network and the authors could consider using additional markers and co-localisation studies.

B) Each reviewer had concerns about the uncertainty underlying mechanisms whereby VCP exerts effects on lysosome morphology and/or function. Does VCP separately regulate lysosome tubulation, the fusion of autophagosomes with lysosomes and/or some other critical function of lysosomes?

We do see that addressing the first set of points (A) appears very feasible in a timely manner. Answers to these queries would likely solidify but probably not grossly change the current set of morphological observations. Dissecting out the issue of mechanism is likely to be beyond what the authors could hope to achieve in a timely manner. While we feel that 'descriptive' findings of the paper are actually of sufficient interest to be published in *eLife* in the absence of further experiments to demonstrate 'mechanisms', we are concerned that the final version of the paper should be careful not to overreach in claiming more mechanistic insight than is supported by the actual data. If the data cannot distinguish between multiple putative lysosomal actions for VCP, then the paper should state and reflect this. The authors have been reasonably careful in this regard, but this is just a flag that the revisions should continue to adhere to this path.

Simply discussing the concerns expressed in (B) above should be sufficient and we do not require experiments in this direction.

Main comments:

The demonstration that lysosomes in *Drosophila* muscles exhibit a very dramatic tubular morphology is unexpected, intriguing and new. The suggestion is that the collapse of this tubular network of lysosomes might represent a mechanism through which impaired valosin containing protein (VCP) function contributes to human disease. This collapse of the lysosomal tubular network is accompanied by defects in autophagosome-lysosome fusion and the accumulation of lipofuscin within lysosomes. VCP is best characterized as an ATPase that facilitates the degradation of ubiquitinated proteins and is vaguely linked to diverse other cellular processes. The authors build on their new observations to suggest a novel role for VCP in promoting either lysosome tubulation, the fusion of autophagosomes with lysosomes and/or some other critical function of lysosomes. While each of these possibilities is reasonable based on the available data, the manuscript ultimately suffers from the lack of a clear elucidation of the specific contribution of VCP to maintenance of lysosome function. Does VCP contribute to one specific aspect of lysosome function that when disrupted leads to the various phenotypes characterized in this study? Or do these phenotypes reflect multiple distinct functions of VCP? While the tubular morphology of lysosomes in muscle represents a striking new observation and the dissemination of this knowledge has great value, however, it is not clear that the disruption of lysosomal tubulation by VCP depletion/mutations underlies the defects in lysosome function (autophagosome fusion and lipofuscin accumulation) that are simultaneously observed under such conditions. Thus, the major limitation is the lack of a precise elucidation of the function of VCP at lysosomes. The authors should keep in mind the comments in the overview, and here, to prioritise how to address this 'main comment'.

Specific points to be addressed:

1) While the images of this lysosomal network are striking and the description of the dynamics is succinct, there is hardly any quantitative statement in their analyses, or in experimental procedures e.g. unclear how many larvae prep were imaged in each experiment. Figure 2: how penetrant of the rescue of VCP(RNAi) by hVCP is? Generally, based on what criteria, a transgene is over-expressed?

2) In the first paragraph of the subsection “VCP is required for the integrity and dynamics of the extended tubular lysosomal network”: The authors state that they pursued a candidate-based RNAi screen, but the list of candidates is missing. Do they want to list them here, or leave it out?

3) It is still somewhat confusing regarding VCP's role in this lysosomal network formation or maintenance. The Phalloidin staining of muscle of VCP-RNAi in Figure 6—figure supplement 1 also seems to show quite noticeable muscle structural defects.

4) Regarding the analyses of the disease-associated mutant VCP, are these mutations loss of function or gain of function? Can they analyse these mutant VCP using the VCP(RNAi) plus hVCP transgene expression scheme as in Figure 2?

5) In the second paragraph of the subsection “*Drosophila* sarcoplasmic lysosomes form an extended dynamic tubular array”, while the lysotracker staining supports the Spin-RFP structure is not an artifact, it remains unclear whether over-expressing Spin causes expansion or distortion of this network as the images in Figure 1 do look different. It would seem that they can resolve this issue by lysotracker staining in Spin-RFP or Spin-GFP lines.

6) In the third paragraph of the subsection “Autophagosome membranes co-localize with the tubular lysosome network” and Figure 3, the GFP-mcherry-Atg8 reporter is very clever. This construct should be referenced and shown the figure. The authors may want to mention that Spin-GFP is not quenched due to topology of the protein.

7) VCP has been previously implicated in the maturation of autophagosomes (Ju et al., JCB, 2009; Tresse et al., Autophagy, 2010). Also in the endosomal sorting of cargos for lysosomal degradation (Ritz et al., NCB, 2011). This new manuscript essentially confirms these results in the *Drosophila* muscle but do not offer further significant insight into mechanism.

8) The specific defect in lysosomes that arises in VCP depleted cells remains vague. Normal cathepsin L maturation and lysotracker labeling suggest that some basic functions are retained. Nonetheless, the accumulation of lipofuscin indicates a defect. More insight into the specific lysosome defect would be helpful in nailing down the actual function of VCP at lysosomes (this might be beyond the scope of the current manuscript, see Overview above).

9) It is not clear why the alterations in mitochondrial morphology in VCP depleted cells might reflect an autophagy defect.

10) The authors identify the tubular network as lysosomal based on the localization of Spinster, a transporter previously identified on lysosomal vesicles, as well as staining of the tubular network with lysotracker (Figure 1). However, to fully confirm that lysosomes are the main (and only) constituents, other markers should be examined, including LAMP1/2 and Cathepsins. They should also show that bona fide Golgi and ER markers do not co-localize with Spin-RFP, and clarify the spatial relationship of these compartments to the tubular network.

11) What is the ultrastructural appearance of this network? Is it electron-dense, as are classical lysosomes? Does it contain multi-vesicular body-like objects? Is its limiting membrane single or double? Although EM-grade fixation may disrupt network architecture, an EM analysis of the resulting vesicles could still yield useful information about their composition and origin. The authors may want to consider whether this experimental avenue can be speedily explored.

12) Staining of the tubular network with lysotracker, a dye that accumulates in acidic compartments, supports that this network is primarily composed of acidic organelles (Figure 1). The vacuolar H+ ATPase (V-ATPase) mediates acidification of lysosomes. Do V-ATPase inhibitors such as concanamycin and Bafilomycin A dissipate the internal acidity of the network? We feel that these experiments should suffice to demonstrate the importance of acidification. However, regarding pH, while lysotracker suggests that this compartment is acidic, it does not allow a precise determination of pH values. Precise measurement of pH may be technically challenging and documenting regional differences not of significant additional value. But, if feasible and the authors wish to embark on this direction, they could employ ratiometric GFP constructs or, alternatively, Lysosensor dyes. By building a calibration curve and interpolating the value, they should be able to come up with a precise value (possibly matching the lysosomal internal pH, which is within the 4.0-5.0 range) and also determine whether pH is homogeneous throughout the network or if subregions with different acidity values exist.

13) The autophagic marker Atg8 is distributed along the length of the tubular network, albeit only in certain subregions (Figure 3). The authors propose that these are regions of ongoing autophagosome-lysosome fusion, leading to progressive Atg8 degradation. One important question is whether this continuous influx of autophagic membranes may in fact enable the accretion of the lysosomal tubular network. The authors should knock down key autophagic mediators such as Atg5 or Atg7, and measure the impact on the morphology of Spns-RFP positive vesicles.

14) Treatment with the VCP inhibitor DbEQ and with the proteasome inhibitor MG132 leads to the formation of VCP-GFP positive rod-like structures and an overall increase in cellular VCP levels (Figure 4). Thus, under normal circumstances VCP may be subjected to rapid degradation. The authors should verify this point by quantifying total VCP-GFP fluorescence as well as by western blotting.

15) Treatment with MG132 causes the dissolution of the tubular network (Figure 4). This effect is attributed to the sequestration of VCP in rod-like structures, which would block its action in a dominant-negative fashion. However, the link here is tenuous given that many proteins undergoing proteasome-regulated turnover could be involved in the tubular network architecture. Another way to test this point is to induce VCP aggregation in a proteasome-independent way using protein homo- or hetero-dimerization systems (i.e. FKBP-FRB) to see whether a similar effect is obtained. If successful, the authors should use this technique to verify that the same phenotypes observed upon RNAi-mediated suppression of VCP (poly-Ub aggregates, mitochondrial damage) also occur.

16) An important question concerns the mechanism of action of VCP in lysosomal network maintenance, autophagosome-lysosome fusion etc. Are these effects due to a direct action of VCP on lysosomal and autophagosomal proteins? Is it an indirect effect of VCP loss of function due, for instance, to ER stress? Does induction of ER stress by other means result in the same effects?

---

## [Author Response]

*Based on the overall evaluation and comments arising from the collective reviews, we first outline the broad areas of concern, which can be binned into two key points*.

*A) The referees agree that the novelty that makes this manuscript potentially suitable for* eLife *is the tubular organisation of the lysosomal network in muscle. A more thorough characterisation of this striking sub-cellular structure will therefore be valuable and necessary. Better quantification and explanation of sample sizes as well as controls are needed to demonstrate that the observation of the lysosomal phenotypes seen are not biased by choice of probes used (Spinster vs. Lysotracker for example). These will much strengthen the characterisation of the cellular network and the authors could consider using additional markers and co-localisation studies*.

We appreciate the reviewers’ comments, which we have addressed with new experiments and text revisions. These changes are briefly listed here and are further detailed in the responses to specific criticisms (below). Collectively, these additional experiments provide a more thorough characterization of the tubules and more clearly define these tubules as lysosomal. Data for items A1-A3 are now reported in new Figure 2.

A1) We performed experiments to co-label the lysosomal tubules with Spinster-GFP and Lysotracker in the same muscle. We show near perfect co-localization between Spinster-GFP and Lysotracker, demonstrating that these probes are marking the same tubular organelle structures.

A2) We have co-imaged Spin-RFP labeled lysosomal tubules with several additional markers including ER tracker, medial and trans golgi markers as well as markers for both early and recycling endosomes. We also performed co-imaging of Spin-RFP with a mitochondrial marker. All of these markers show distinct localization patterns compared to Spin-RFP labeled tubules. As a note to the reviewers, golgi organization in skeletal muscle is distinct from the classical golgi stacks observed in many cells. Vertebrate muscle shows a unique, distributed, vesicular organization of the golgi throughout the muscle (Ralston et al., J. Neurosci., 2001). We find that this organization is conserved in *Drosophila* muscle.

A3) We performed an additional experiment to test the acidification of Spin-labeled tubules. We treated muscles with the V-ATPase specific inhibitor concanamycin A and observed a decrease in acidity in the tubules. ConA inhibits the lysosomal V-ATPase, verifying that the Spin-positive tubules are acidic and lysosomal.

A4) We have modified the text to clarify sample sizes and our approach to the presentation of data. Regarding the issue of sample size. Tubules are present in every muscle that we have examined. Thus, phenotypic penetrance is essentially 100%. In addition, we went to the extent of imaging these tubules in the live, un-dissected animal (an experiment reported in the original submission). This was done to verify that the presence of tubules is not an artifact of animal dissection. We see tubules in all muscle cells in the live organism. We have added a statement to the text to this effect (Results section): “Tubules were observed in every muscle and there were no apparent differences in tubule abundance or architecture between different muscles.” We have included samples sizes in figure legends where necessary to interpret statistical comparisons. We also now state in the Methods section that at least 3 muscles in 3 animals (N≥9) were imaged and the most representative images are shown (subsection “Microscopy methods”). Regarding the collection of live imaging data for movies, more than 10 muscles were imaged (N≥10) for each experiment. In most instances, many more muscles than this were imaged for our experiments.

*B) Each reviewer had concerns about the uncertainty underlying mechanisms whereby VCP exerts effects on lysosome morphology and/or function. Does VCP separately regulate lysosome tubulation, the fusion of autophagosomes with lysosomes and/or some other critical function of lysosomes*?

*We do see that addressing the first set of points (A) appears very feasible in a timely manner. Answers to these queries would likely solidify but probably not grossly change the current set of morphological observations. Dissecting out the issue of mechanism is likely to be beyond what the authors could hope to achieve in a timely manner. While we feel that 'descriptive' findings of the paper are actually of sufficient interest to be published in* eLife *in the absence of further experiments to demonstrate 'mechanisms', we are concerned that the final version of the paper should be careful not to overreach in claiming more mechanistic insight than is supported by the actual data. If the data cannot distinguish between multiple putative lysosomal actions for VCP, then the paper should state and reflect this. The authors have been reasonably careful in this regard, but this is just a flag that the revisions should continue to adhere to this path*.

*Simply discussing the concerns expressed in (B) above should be sufficient and we do not require experiments in this direction*.

We thank the reviewers for their consideration and for their enthusiasm regarding our basic findings. Indeed, we hope that our work will spur new ideas and experimentation on VCP activity in muscle, which is one of the primary tissues compromised in human patients harboring disease-causing mutations in VCP. We have made some changes to the text, particularly in the Discussion, to make sure that we are appropriately circumspect regarding the mechanism of action of VCP.

*Main comments*:

*The demonstration that lysosomes in* Drosophila *muscles exhibit a very dramatic tubular morphology, is unexpected, intriguing and new. The suggestion is that the collapse of this tubular network of lysosomes might represent a mechanism through which impaired valosin containing protein (VCP) function contributes to human disease. This collapse of the lysosomal tubular network is accompanied by defects in autophagosome-lysosome fusion and the accumulation of lipofuscin within lysosomes. VCP is best characterized as an ATPase that facilitates the degradation of ubiquitinated proteins and is vaguely linked to diverse other cellular processes. The authors build on their new observations to suggest a novel role for VCP in promoting either lysosome tubulation, the fusion of autophagosomes with lysosomes and/or some other critical function of lysosomes. While each of these possibilities is reasonable based on the available data, the manuscript ultimately suffers from the lack of a clear elucidation of the specific contribution of VCP to maintenance of lysosome function. Does VCP contribute to one specific aspect of lysosome function that when disrupted leads to the various phenotypes characterized in this study? Or do these phenotypes reflect multiple distinct functions of VCP? While the tubular morphology of lysosomes in muscle represents a striking new observation and the dissemination of this knowledge has great value, however, it is not clear that the disruption of lysosomal tubulation by VCP depletion/mutations underlies the defects in lysosome function (autophagosome fusion and lipofuscin accumulation) that are simultaneously observed under such conditions. Thus, the major limitation is the lack of a precise elucidation of the function of VCP at lysosomes. The authors should keep in mind the comments in the overview, and here, to prioritise how to address this 'main comment'*.

We agree that we have not pinpointed the specific activity of VCP that controls the integrity of the lysosomal tubule network in muscle. We thank the reviewers for acknowledging that doing so will be a major effort for the future, given the diverse functions that have been ascribed to VCP in the cell. Although the experiments required to elucidate the specific contributions of VCP on lysosome tubulation are beyond the scope of the present study, our findings expose a new avenue of VCP biology that is likely to spur many more studies in this direction. None-the-less, in response to this criticism, we have strengthened our treatment of VCP in our Discussion section. We now state, “VCP and the yeast homologue Cdc48 have been ascribed many functions within the cell including cell cycle progression (28), UPS and ERAD protein degradation (26; 48), mitophagy (40) and classical autophagy (7; 19; 26; 41)… To this list of VCP-mediated activities we now add the action of VCP in controlling the integrity and dynamics of a tubular lysosomal system and fusion of autophagosomes with tubular lysosomes. Additional, future experimentation will be necessary to determine which specific VCP-mediated molecular mechanism(s) is most directly relevant to the integrity and dynamics of lysosomal tubules in muscle” (subsection “Consequences of tubular lysosome dysfunction”).

*Specific points to be addressed*:

*1) While the images of this lysosomal network are striking and the description of the dynamics is succinct, there is hardly any quantitative statement in their analyses, or in experimental procedures e.g. unclear how many larvae prep were imaged in each experiment.*
Figure 2*: how penetrant of the rescue of VCP(RNAi) by hVCP is? Generally, based on what criteria, a transgene is over-expressed*?

We have modified the text to clarify sample sizes and our approach to the presentation of data. Regarding the issue of sample size. Tubules are present in every muscle that we have examined. Thus, phenotypic penetrance is essentially 100%. In addition, we went to the extent of imaging these tubules in the live, un-dissected animal (an experiment reported in the original submission). This was done to verify that the presence of tubules is not an artifact of animal dissection (please see point A4 above).

For rescue experiments, the phenotypic characterization is all or none; the tubules are absent in the knockdown and present in the rescue muscle. The phenotype was scored in N≥10 rescue animals. We did not attempt to make any quantitative estimate of the magnitude of the rescue as the phenotype was an all or none assessment with ∼100% penetrance. We modified the results to say “Over-expressing human *VCP* in *dVCP*^*RNAi*^ muscles rescued the formation of lysosome tubules in every muscle” to clarify (subsection “VCP is required for the integrity and dynamics of the extended tubular lysosomal network”).

*2) In the first paragraph of the subsection “VCP is required for the integrity and dynamics of the extended tubular lysosomal network”: The authors state that they pursued a candidate-based RNAi screen, but the list of candidates is missing. Do they want to list them here, or leave it out*?

This screen is still ongoing. As such, we would prefer to leave out the identification of the other hits that we are finding. We thank the editors for their understanding.

*3) It is still somewhat confusing regarding VCP's role in this lysosomal network formation or maintenance. The Phalloidin staining of muscle of VCP-RNAi in*
Figure 6—figure supplement 1
*also seems to show quite noticeable muscle structural defects*.

Based on our results, we cannot distinguish between VCP’s role in formation or maintenance and we have added a sentence in the Discussion to make this point more clear (“Although we cannot distinguish the role of VCP in the initiation and/or maintenance of lysosome tubules, these data argue that VCP acts at the autolysosmal membrane to control autolysosomal dynamics and the progression of autophagic protein clearance”). We agree that the muscles are visibly degrading in VCP RNAi animals. But, other degenerative mutants (Figure 3) do not cause disruption of the lysosome network, indicating that the loss of the tubular network is not merely a byproduct of muscle degeneration and is a specific defect of VCP loss. This is stated in the Results section (subsection “VCP is required for the integrity and dynamics of the extended tubular lysosomal network”).

*4) Regarding the analyses of the disease-associated mutant VCP, are these mutations loss of function or gain of function? Can they analyse these mutant VCP using the VCP(RNAi) plus hVCP transgene expression scheme as in*
Figure 2?

VCP forms a hexamer and over-expressing the disease mutations in the context of wildtype VCP produces a dominant-negative effect. It is not clear whether over-expression of human-VCP would provide much additional information and we have not pursued this further. However, given that VCP RNAi produces a phenotype that is similar to the over-expression of mutant VCP, our results imply that the VCP mutants have a loss-of-function effect on lysosomal tubule integrity. This point has been added to the end of the Results section.

*5) In the second paragraph of the subsection “*Drosophila *sarcoplasmic lysosomes form an extended dynamic tubular array”, while the lysotracker staining supports the Spin-RFP structure is not an artifact, it remains unclear whether over-expressing Spin causes expansion or distortion of this network as the images in*
Figure 1
*do look different. It would seem that they can resolve this issue by lysotracker staining in Spin-RFP or Spin-GFP lines*.

It is our impression that the apparent difference between Lysotracker staining and Spin-RFP (or Spin-GFP) is caused by the intensity of the different labels; a membrane-localized RFP versus membrane permeable dye. It remains possible that there is a distorting effect of Spin-RFP or –GFP on the tubule network, but this seems tangential to the main line of investigation for this paper. We have now shown that Lysotracker co-localizes with Spin-GFP, so we are certain that these reagents label the same compartment. These data are now included in new Figure 2.

*6) In the third paragraph of the subsection “Autophagosome membranes co-localize with the tubular lysosome network” and*
Figure 3*, the GFP-mcherry-Atg8 reporter is very clever. This construct should be referenced and shown the figure. The authors may want to mention that Spin-GFP is not quenched due to topology of the protein*.

We have added the reference for the GFP-mCherry-Atg8a reporter in the text (subsection “Autophagosome membranes co-localize with the tubular lysosome network”). We have also added a sentence and reference to clarify that Spin-GFP is not quenched because the C-terminal GFP tag resides on the cytoplasmic side of the lysosome membrane.

*7) VCP has been previously implicated in the maturation of autophagosomes (Ju et al., JCB, 2009; Tresse et al., Autophagy, 2010). Also in the endosomal sorting of cargos for lysosomal degradation (Ritz et al., NCB, 2011). This new manuscript essentially confirms these results in the Drosophila muscle but do not offer further significant insight into mechanism*.

As stated above, we have added new text to our Discussion acknowledging that it will take considerable future experimentation to pinpoint the specific activities of VCP that control the integrity of the lysosomal tubule network in muscle. I believe that we have also been careful to acknowledge and cite relevant previous studies including [19] and Tresse et al., (2010). We are grateful to the editors for acknowledging that we also make an experimental advance through the identification and characterization of a tubular lysosomal network in muscle that requires VCP and is influenced by disease-causing mutations in VCP.

*8) The specific defect in lysosomes that arises in VCP depleted cells remains vague. Normal cathepsin L maturation and lysotracker labeling suggest that some basic functions are retained. Nonetheless, the accumulation of lipofuscin indicates a defect. More insight into the specific lysosome defect would be helpful in nailing down the actual function of VCP at lysosomes (this might be beyond the scope of the current manuscript, see Overview above)*.

We believe that it is beyond the scope of this study and thank the editors for acknowledging that this might be the case.

*9) It is not clear why the alterations in mitochondrial morphology in VCP depleted cells might reflect an autophagy defect*.

Damaged mitochondria commonly exhibit altered morphology and membrane potential. Because damaged mitochondria are typically cleared though mitophagy, the accumulation of mitochondria with altered morphology suggests a defect in mitochondrial clearance. We have included additional text in the corresponding section of the results to clarify this point (“The swollen mitochondria observed in the VCP RNAi expressing muscles are likely defective and should be a prime target for mitophagy-dependent degradation”).

*10) The authors identify the tubular network as lysosomal based on the localization of Spinster, a transporter previously identified on lysosomal vesicles, as well as staining of the tubular network with lysotracker (*Figure 1*). However, to fully confirm that lysosomes are the main (and only) constituents, other markers should be examined, including LAMP1/2 and Cathepsins. They should also show that bona fide Golgi and ER markers do not co-localize with Spin-RFP, and clarify the spatial relationship of these compartments to the tubular network*.

We have done our best to address the reviewers’ comments with available reagents and have added an entire new figure of data to our manuscript (new Figure 2). First, we have performed several of the suggested experiments. We co-imaged Spin-RFP with several other previously characterized organelle markers. Co-imaging Spin-RFP with previously characterized markers of mitochondria (mito tracker), ER (ER tracker), medial golgi (ManII), trans golgi (galT), early endosomes (Rab5) and recycling endosomes (Rab11) clearly demonstrates that Spin-RFP labeling is distinct from all of these organelles. These data are now presented in a new figure (Figure 2). We thank the reviewers for prompting us to pursue these experiments. These data significantly extend our understanding of the tubular lysosomal network within the complex muscle sarcoplasm. As a note to the reviewers, golgi organization in skeletal muscle is distinct from the classical golgi stacks observed in many cells. Vertebrate muscle shows a unique, distributed, vesicular organization of the golgi throughout the muscle (Ralston et al., J. Neurosci., 2001). Our data show that this organization is conserved in *Drosophila* muscle.

Unfortunately, we were unable to perform some of the other experiments suggested by the reviewers. First, to our knowledge, fluorescent probes for Cathepsins in *Drosophila* do not exist and the process of generating new reagents and verifying their activity would be beyond the stated scope of *eLife* reviews. We also attempted to co-localize Spin-RFP with Lamp1-GFP, but found that the currently available Lamp1-GFP constructs do not express well in muscles using either of two different muscle specific Gal4 drivers (MHC-Gal4 or BG57-Gal4). However, the reviewers should be aware that when Spin is expressed in HeLa cells, Spin-GFP co-localizes with Lamp1 (Sweeney et al., 2001). Furthermore, we show strong co-localization between muscle expressed Spin-RFP and Lysotracker (Figure 2). We also provide new evidence that the lysosomal vATPase is necessary to maintain tubule acidification, consistent with lysosomal identify. When taken into consideration with additional information described below, there is considerable evidence that the tubular network in muscle is lysosomal.

*11) What is the ultrastructural appearance of this network? Is it electron-dense, as are classical lysosomes? Does it contain multi-vesicular body-like objects? Is its limiting membrane single or double? Although EM-grade fixation may disrupt network architecture, an EM analysis of the resulting vesicles could still yield useful information about their composition and origin. The authors may want to consider whether this experimental avenue can be speedily explored*.

These are all excellent questions (electron density, limiting membrane, etc.). We have performed extensive TEM on *Drosophila* muscle. But, without an independent electron-dense marker (immuno-gold or HRP) there is no way for us to be sure about the identification of these lysosomal structures. Working out the details by which we can unambiguously identify lysosomes in muscle is beyond the scope of the current work, but something that we wish to pursue in future studies.

*12) Staining of the tubular network with lysotracker, a dye that accumulates in acidic compartments, supports that this network is primarily composed of acidic organelles (*Figure 1*). The vacuolar H+ ATPase (V-ATPase) mediates acidification of lysosomes. Do V-ATPase inhibitors such as concanamycin and Bafilomycin A dissipate the internal acidity of the network? We feel that these experiments should suffice to demonstrate the importance of acidification. However, regarding pH, while lysotracker suggests that this compartment is acidic, it does not allow a precise determination of pH values. Precise measurement of pH may be technically challenging and documenting regional differences not of significant additional value. But, if feasible and the authors wish to embark on this direction, they could employ ratiometric GFP constructs or, alternatively, Lysosensor dyes. By building a calibration curve and interpolating the value, they should be able to come up with a precise value (possibly matching the lysosomal internal pH, which is within the 4.0-5.0 range) and also determine whether pH is homogeneous throughout the network or if subregions with different acidity values exist*.

We have examined the effect of ConA treatment on GFP-mCherry-Atg8a fluorescence and localization in muscles. Under basal conditions, GFP-mCherry-Atg8a localizes to tubules, but the GFP fluorescence is quenched inside the tubules due to the acidic environment and only mCherry fluorescence is observed. However, upon treatment of ConA for 3 hours, we observed GFP and mCherry tubule labeling. These data reveal several important aspects of the lysosome tubules. First, it verifies by another method that the tubules are indeed acidic. Second, it indicates that the acidity of the tubules is dependent upon the V-ATPase, a lysosomal-specific proton pump. Finally, these data show that acidification of the lysosomes is not required to maintain the tubular architecture. These new data have been added to Figure 5.

We agree that quantitative exploration of the pH within the lysosomal network and the possibility that regional differences exist would be very interesting. However, we believe that it is beyond the scope of the current work.

*13) The autophagic marker Atg8 is distributed along the length of the tubular network, albeit only in certain subregions (*Figure 3*). The authors propose that these are regions of ongoing autophagosome-lysosome fusion, leading to progressive Atg8 degradation. One important question is whether this continuous influx of autophagic membranes may in fact enable the accretion of the lysosomal tubular network. The authors should knock down key autophagic mediators such as Atg5 or Atg7, and measure the impact on the morphology of Spns-RFP positive vesicles*.

We have done the suggested experiment and find that inhibition of Atg7 by RNAi (a previously characterized RNAi line) does not disrupt the lysosome tubular network. Thus, continuous autophagic flux is not necessary for the integrity of the lysosomal network. These data have been added to Figure 4.

*14) Treatment with the VCP inhibitor DbEQ and with the proteasome inhibitor MG132 leads to the formation of VCP-GFP positive rod-like structures and an overall increase in cellular VCP levels (*Figure 4*). Thus, under normal circumstances VCP may be subjected to rapid degradation. The authors should verify this point by quantifying total VCP-GFP fluorescence as well as by western blotting*.

We would like to clarify that we do not think cellular VCP levels increase upon treatment with MG132 or DBeQ. This is based on our observation that upon washout of MG132, the VCP rods dissolve and the cytoplasmic pool of VCP is restored to resting state. None-the-less, we followed the reviewers’ suggestion and examined total VCP levels upon treatment with MG132 or DBeQ by western blot analysis and did not observe any increase in total VCP levels. We have added these data to Figure 5.

*15) Treatment with MG132 causes the dissolution of the tubular network (*Figure 4*). This effect is attributed to the sequestration of VCP in rod-like structures, which would block its action in a dominant-negative fashion. However, the link here is tenuous given that many proteins undergoing proteasome-regulated turnover could be involved in the tubular network architecture. Another way to test this point is to induce VCP aggregation in a proteasome-independent way using protein homo- or hetero-dimerization systems (i.e. FKBP-FRB) to see whether a similar effect is obtained. If successful, the authors should use this technique to verify that the same phenotypes observed upon RNAi-mediated suppression of VCP (poly-Ub aggregates, mitochondrial damage) also occur*.

Certainly, proteasome inhibition has many effects on the cell. However, we want to emphasize that the point of this experiment was not to implicate proteasome activity in the maintenance of tubular network architecture. Rather, the discovery that MG132 caused VCP aggregation was a fortuitous observation that allowed us to do a *recovery* experiment. We used MG132 to induce sequestration of VCP into cytoplasmic aggregates. Then, we were able to wash out MG132 and watch the effect(s) of re-establishing a cytoplasmic pool of soluble VCP. It is clear from the imaging data that VCP is sequestered from the cytoplasm during VCP aggregation and that cytoplasmic VCP is restored upon MG132 washout and tubulation was restored.

*16) An important question concerns the mechanism of action of VCP in lysosomal network maintenance, autophagosome-lysosome fusion etc. Are these effects due to a direct action of VCP on lysosomal and autophagosomal proteins? Is it an indirect effect of VCP loss of function due, for instance, to ER stress? Does induction of ER stress by other means result in the same effects*?

We have done the suggested experiment to test for the effects of ER stress. We induced ER stress by treating muscles with Tunicamycin (4 hours) and saw no effect on the integrity of the tubular lysosomal network. We confirmed the induction of ER stress by extracting protein from the same preps that were imaged and analyzing Hsc-70/BiP (a protein known to be induced by ER stress) levels by western blot. These data are included in new Figure 3.